# Secure Multi-agent Reinforcement Learning for Service Systems with Affinity and Byzantine Nodes: Stability Analysis and Protection Design

Yifan Jiang [* 1]  Jiasheng Pan [* 1]  Mengtian Li [2]  Li Jin [1]

## Abstract

We study decentralized multi-agent reinforcement learning (MARL) for networked service systems with affinity in the presence of Byzantine nodes. The way that a server processes a job depends on an affinity state that captures the correlation between the job and the server. Each node learns a local control policy via an actor-critic algorithm with linear function approximation over inherently unbounded space of traffic states, while exchanging parameter information with neighbors through a communication graph. A set of Byzantine agents can exploit the unbounded state space to compromise the consensus mechanism, destabilizing both learning and queuing processes. To address this vulnerability, we propose a resilient consensus-based MARL algorithm, which mitigates adversarial parameter manipulation and guarantees traffic stability under mild assumptions. We prove that the cooperative agents' policies converge almost surely to a bounded neighborhood of a stationary solution of the global objective. We demonstrate the effectiveness and generality of the proposed framework in several representative service systems, including semantic routing for large language model serving, distributed polling in cloud computing, and smart manufacturing logistics.

## 1. Introduction

### 1.1. Motivation

In many service systems, how a server processes a job depends on their mutual affinity. For instance, a graphics processing unit (GPU) typically handles requests more efficiently when they align with data already stored in the GPU's key-value (KV) cache (Kim & Song, 2023). Similar dynamics are observed in edge computing (Ahmed & Ahmed, 2016) and smart manufacturing (Monostori et al., 2016). Such job-server affinity often leads to distributed control architectures, for which multi-agent reinforcement learning (MARL) is increasingly popular (Zhang et al., 2018). However, such decentralized algorithms are susceptible to Byzantine agents, which can disrupt convergence by disseminating malicious information and causing job queues and/or training error to diverge (Wu et al., 2021).

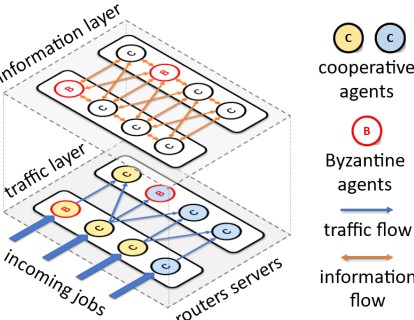

*Figure 1.* The illustrative service system with 4 routers and 4 servers has 2 Byzantine nodes. Router-server pairs with high affinities are illustrated with the same color in the traffic layer, so a router-server pair with matching (resp. non-matching) colors is associated with high (resp. low) service rate. Byzantine agents may disseminate malicious information in the information layer to compromise performance of the traffic layer.

In this paper, we study MARL for a class of distributed service systems with affinities in the face of Byzantine nodes (Figure 1). Specifically, we focus on three objectives: (1) to characterize how Byzantine agents may destabilize both traffic state and learning process; (2) to design defensively robust strategies with provable stability guarantees; and (3) to demonstrate the applicability of our framework in practical cases such as large language model (LLM), multi-agent edge computing (MEC), and smart manufacturing.

### 1.2. Related work

MARL is widely used in large-scale service systems as a scalable control paradigm (Zhang et al., 2018; Lin et al., 2021; Qu et al., 2022; Huh & Mohapatra, 2024). In par-

---
[*]Equal contribution [1]Global College, Shanghai Jiao Tong University [2]Shanghai Film Academy, Shanghai University. Correspondence to: Li Jin <li.jin@sjtu.edu.cn>.

*Proceedings of the 43rd International Conference on Machine Learning*, Seoul, South Korea. PMLR 306, 2026. Copyright 2026 by the author(s).

ticular, consensus-based actor-critic algorithms have been shown to converge under standard assumptions (Zhang et al., 2018). Extensions to nonlinear approximation and partial information sharing have further improved applicability to large-scale systems (Qu et al., 2019; Gabler & Wollherr, 2024). However, existing resilient MARL methods generally assume bounded state spaces or bounded basis functions (Wu et al., 2021; Medhi et al., 2023; Li et al., 2023), which do not directly address the risk of divergence. Furthermore, the integration of job-server affinities has not been well studied.

The impact of Byzantine agents has been extensively studied in distributed optimization and learning. Robust aggregation rules such as trimmed-mean, median-based methods, and Krum (Chen et al., 2017; Blanchard et al., 2017; Yin et al., 2018) have been proposed to mitigate the influence of adversarial updates in stochastic gradient descent. These ideas have recently been adopted in consensus-based MARL (Lin et al., 2021; Wu et al., 2023) to provide resilience. Nevertheless, the impact of Byzantine agents on learning updates driven by unbounded state-dependent gradients has not been fully understood.

Reinforcement learning has been applied to a variety of service systems unbounded state spaces (Singh & Bertsekas, 1996; Mao et al., 2016). Classical results emphasize stability and throughput optimality in systems with unbounded state spaces (Tassiulas & Ephremides, 1990). Recent works have explored actor-critic and policy-gradient methods (Jain et al., 2024; Tian et al., 2026; Chen et al., 2020a). These general-purpose results do not directly apply to our problem but provide insights for our analysis.

### 1.3. Our contributions

We propose a unified modeling framework for decentralized control in service networks with inherently unbounded state spaces and job-server affinities, in the face of Byzantine agents. The system is formulated as a directed service network consisting of routers and servers, where jobs propagate through queues with infinite buffers. Beyond queue backlogs, we introduce a time-varying *affinity state* that captures heterogeneous compatibility between jobs and servers. Each node acts as a learning agent making local decisions. The overall dynamics are modeled as a decentralized multi-agent Markov decision process (MMDP). We further incorporate a Byzantine threat model in which a subset of agents can arbitrarily manipulate their exchanged information, directly interfering with learning and consensus.

Based on the above formulation, we design a resilient MARL algorithm with synchronized policy mixture. Our algorithm relies on linear function approximation (LFA) because of their consistency with service system dynamics (Chen et al., 2020b) as well as good theoretical properties

and computational efficiency compared with deep reinforcement learning. Byzantine resilience is achieved by applying consensus directly to the learning parameters, while the synchronized policy mixture plays a complementary role in regulating the learning-induced state distribution. In particular, by mixing the updated policy with a common reference policy across agents, the mechanism effectively regularizes policy evolution and prevents abrupt behavioral shifts that may destabilize queue dynamics or induce state explosion in unbounded service systems. Consequently, this integrated design yields more stable training iterations and service processes, and leads to stronger empirical performance, especially in environments with unbounded state spaces. We further prove that the proposed algorithm converges almost surely under mild assumptions.

We instantiate the proposed framework in several representative networked systems that exhibit unbounded state dynamics and decentralized control. These include large language model semantic routing, multi-agent edge computing, and smart manufacturing delivery; see Figure D. These case studies demonstrate the generality of the proposed modeling framework and numerically validate the stabilizing and resilience benefits of the proposed algorithm in realistic network settings. In particular, we show how Byzantine agents may mislead cooperative agents to make low-affinity decisions, and how the proposed algorithm mitigates such misbehavior.

## 2. Model and Algorithm

### 2.1. Service system with affinity

For the service system with the structure in Figure 1, we consider a multi-agent Markov decision process (MMDP) characterized by a tuple $\mathcal{M} = \langle \mathcal{G}, \mathcal{S}, \mathcal{A}, P, r, \gamma \rangle$ defined as follows.

The directed graph $\mathcal{G} = \langle \mathcal{N}, \mathcal{E} \rangle$ specifies the topology of the system, where $\mathcal{N}$ and $\mathcal{E}$ are the set of nodes and the set of links, respectively. The nodes can be partitioned into two categories, viz. routers $\mathcal{N}_r$ and servers $\mathcal{N}_s$. A router $\mathcal{N}_r$ receives jobs from outside and allocates jobs to servers. A server $\mathcal{N}_s$ receives and serves jobs from routers. Let $\text{In}(i)$ (resp. $\text{Out}(i)$) be the set of upstream (resp. downstream) neighbors to node $i$. Let $m$ be the cardinality of $\mathcal{E}$.

The traffic state of the system is $x(t) = [x_1(t), x_2(t), \ldots, x_n(t)]^T \in \mathbb{R}_{\geq 0}^N$, where $N$ is the cardinality of $\mathcal{N}$. The state $x_i(t)$ of a router $i \in \mathcal{N}_r$ is the number of jobs to be allocated. Jobs enter a router $i$ as a Bernoulli arrival process with rate $\lambda_i \in [0, 1]$. For ease of presentation, we assume that a router can allocate at most one job per step. When a job moves from a router $i$ to a server $j$, the job queue $x_j(t)$ is increased by a random variable $U$, which is the service time for this job. The mean

*(a)* Large language model semantic routing     *(b)* Multi-agent edge computing polling     *(c)* Smart manufacturing delivery

*Figure 2.* Three practical cases of service systems with job-server affinities. Figure 2a: the response time for an incoming request depends on the extent to which the request content matches the KV cache contents of a GPU. Figure 2b: the computation time for an incoming job depends on the type of jobs that the server is processing. Figure 2c: the service time for a task depends on the physical distance between a robot and the task. These systems are usually distributed and susceptible to Byzantine agents.

$u(f_{ij})$ of $U$ is a non-increasing real-valued function of the *affinity* $f_{ij} \in [0, 1]$ between a source (router) and a server. Intuitively, an affinity $f_{ij}$ close to 1 (resp. 0) means that server $j$ is in a very suitable (resp. unsuitable) condition to accommodate jobs from router $i$. Hence, $x_i(t) \in \mathbb{Z}_{\geq 0}$ for a router $i \in \mathcal{N}_{\mathrm{r}}$, while $x_i(t) \in \mathbb{R}_{\geq 0}$ for a server $i \in \mathcal{N}_{\mathrm{s}}$. We are particularly interested in the scenario where $f_{ij}(t)$ is time-varying. Hence, we call the vector $f(t)$ of $f_{ij}(t)$'s the affinity state. Therefore, $f(t) \in [0, 1]^m$. Thus, the state space for the service system is $\mathcal{S} = (\mathbb{R}_{\geq 0}^N \times [0, 1]^m)$, and the state of the service system is $s(t) = (x(t), f(t))$.

The action $a_i$ for a router $i$ is to allocate incoming jobs to downstream nodes; hence $a_i \in \mathrm{Out}(i)$ for each $i \in \mathcal{R}$. The action $a_i$ for a server $i$ is to accept incoming jobs to upstream nodes; hence $a_i \in \mathrm{In}(i)$ for each $i \in \mathcal{T}$. Since every node may take actions that influence the system state, we also call a node an *agent*. We use the unified notation $\mathcal{A}_i$ to denote the action space for $i \in \mathcal{N}$. We also denote the actions for all agents as $a \in \mathcal{A}$, where $\mathcal{A} = \prod_{i \in \mathcal{N}} \mathcal{A}_i$.

We assume that the dynamics of the service system is Markovian and can be characterized by a transition kernel $P((s, a), \cdot)$ such that

$$P((s, a), \mathcal{B}) = \Pr\{s(t + 1) \in \mathcal{B} | s(t) = s, a(t) = a\}$$

for each measurable set $\mathcal{B} \subset (\mathbb{Z}_{\geq 0}^N \times [0, 1]^m)$. The kernel mainly captures the conservation law for the job flows and the job-server correlation captured by the affinity state.

The above generic formulation applies to the following practical cases, which discussed in details in Section 4:

1. Large language model semantic routing (Figure 2a): A set of routers allocate incoming requests (jobs) to a set of parallel GPUs (servers). Each GPU has a memory of the recently served jobs in its KV cache. The affinity captures the how well an incoming request matches the memory in the KV cache and thus influences service performance. The routers are the agents of interest, which adaptively selects the routing actions.

2. Distributed routing for edge computing (Figure 2b): A set of parallel servers adaptively decide the service priority for each source. If a server changes its priority ranking, a switch-over cost is induced. The affinity captures the current priority of a server and influences the switch-over cost. The servers are the agents of interest.

3. Autonomous delivery in smart manufacturing (Figure 2c): A set of physically distributed stations (sources) randomly generate delivery tasks (jobs). A set of robots (servers) selects the tasks to accept. To complete a task, a delivery robot has to pick up the item at the station and deliver to its destination. The affinity captures the physical distance between a station and a robot. The robots are the agents of interest.

$r$ and $\gamma$ are associated with the agent models will be introduced in the next subsection.

## 2.2. Cooperative and Byzantine agents

To study the resiliency of the system in the face misbehaving agents, we also partition the set of agents $\mathcal{N}$ into a set of cooperative agents $\mathcal{N}^+$ and a set of Byzantine agents $\mathcal{N}^-$. The identities and number $b = |\mathcal{N}^-|$ of Byzantine agents are unknown to the cooperative agents but known to the Byzantine agents. We assume a local upper bound $H$ on the number of Byzantine neighbors. To guarantee that the cooperative agents can reach an agreement on global parameters, the underlying communication graph is assumed to be $(2H + 1)$-robust (LeBlanc et al., 2013). This implies that the cooperative subgraph remains sufficiently connected to facilitate information flow, even when each agent executes local outlier trimming strategies.

The one-step cost for a cooperative agent $i \in \mathcal{N}^+$ is specified by a function $r_i : \mathcal{S} \times \mathcal{A}_i \to \mathbb{R}$. A typical choice for $r_i$ is the total workload $r_i(s, a) = \|x\|_1$. Let $r = [r_1, r_2, \ldots, r_n]^T$. The team-average reward for cooper-

ative agents is:

$$r^+(s,a) = \frac{1}{|\mathcal{N}^+|} \sum_{i \in \mathcal{N}^+} r_i(s, a_i). \qquad (1)$$

We will specify how Byzantine agents can compromise the system in Section 2.3.

The cooperative objective is to maximize the return:

$$J^+(\pi^+, \pi^-) \triangleq \mathbb{E}_\pi \left[ \frac{1}{|\mathcal{N}^+|} \sum_{i \in \mathcal{N}^+} \sum_{t=0}^{\infty} \gamma^t r^i(s_t, a_t) \right], \quad (2)$$

where $\gamma \in (0, 1)$ is a discount rate; the expectation is taken with respect to the invariant probability measure of the state $s$ associated with $\pi$.

We consider the scenario where agents select their actions in a decentralized manner. Regarding the information structure, we assume that the global state $s$ is observable to all agents, whereas the reward signals are private; that is, each agent $i$ observes only its local reward $r_i$ and cannot directly access the rewards of others. Consequently, the joint policy of cooperative agents is defined as $\pi^+(a^+ \mid s) = \prod_{i \in \mathcal{N}^+} \pi_i(a_i \mid s)$ and that of Byzantine nodes is $\pi^-(a^- \mid s) = \prod_{i \in \mathcal{N}^-} \pi_i(a_i \mid s)$. The global policy is $\pi = (\pi^+, \pi^-)$. We use $P_\pi(s, \cdot)$ to denote the transition kernel for the MMDP under a fixed policy $\pi$. By (Meyn & Tweedie, 2012), $P_\pi$ can also be used as an operator such that for any measurable function $f$ on $\mathcal{S}$, $P_\pi f(s) = \int_\mathcal{S} P_\pi(s, ds') f(s')$.

## 2.3. Secure learning algorithm

We adopt a decentralized actor-critic framework to solve the cooperative optimization problem above. Each agent $i$ parameterizes its local policy $\pi_i(\cdot|s; \theta_i)$ with a parameter vector $\theta_i$. The primary objective is to update these parameters to maximize the joint return $J^+$. Following the policy gradient theorem, the update direction for agent $i$ is proportional to the gradient of the log-probability weighted by the global advantage function $A^\pi$:

$$\nabla_{\theta_i} J^+ \propto \mathbb{E}_\pi \left[ \nabla_{\theta_i} \log \pi_i(a_i|s; \theta_i) A^\pi(s, a) \right]. \quad (3)$$

In our setting, the advantage function is estimated via the temporal-difference (TD) error relative to the global mean reward:

$$A^\pi(s_t, a_t) \approx \bar{r}_t + \gamma V^\pi(s_{t+1}) - V^\pi(s_t), \qquad (4)$$

where $V^\pi$ denotes the global value function and $\bar{r}_t$ represents the instantaneous global average reward.

We study a class of decentralized actor-critic (AC) algorithms for cooperative agents to compute $\pi^+$. Following standard theoretical analysis (Konda & Tsitsiklis, 1999), we

employ linear approximations for the value function, reward function, and policies. Specifically, each agent $i$ maintains local parameter vectors $v_i$ and $\omega_i$ to approximate the value function and the global average reward, respectively:

$$V(s; v_i) = \phi(s)^\top v_i,$$
$$\bar{r}(s, a; \omega_i) = \psi(s, a)^\top \omega_i, \qquad (5)$$

where $\phi(s)$ and $\psi(s, a)$ are feature vectors.

To evaluate the policy $\pi$, the critic aims to minimize the mean-squared errors (MSE) for both approximations. The corresponding objective functions are defined as:

$$J_\omega(\omega) = \mathbb{E}_\pi \left[ \left( \bar{r}(s, a) - \psi(s, a)^\top \omega \right)^2 \right], \qquad (6a)$$

$$J_v(v) = \mathbb{E}_\pi \Big[ \Big( r(s, a) - \bar{r}(s, a) + \gamma \mathbb{E}[V(s')|s, a]$$
$$- \phi(s)^\top v \Big)^2 \Big], \qquad (6b)$$

where expectations are taken with respect to the stationary distribution induced by $\pi$.

To minimize these objectives, agents perform stochastic semi-gradient descent. We incorporate eligibility traces $z_{i,t}$ to accelerate credit assignment. At each step $t$, agent $i$ computes the local temporal-difference (TD) errors based on its private reward $r_{i,t}$:

$$\delta_{i,t}^v = r_{i,t} + \gamma \phi(s_{t+1})^\top v_{i,t} - \phi(s_t)^\top v_{i,t},$$
$$\delta_{i,t}^\omega = r_{i,t} - \psi(s_t, a_t)^\top \omega_{i,t}. \qquad (7)$$

The trace is updated as $z_{i,t+1} = \gamma \lambda z_{i,t} + \phi(s_t)$. Each agent then performs the **intermediate local update**:

$$\tilde{v}_{i,t+1} = v_{i,t} + \alpha_{v,t} \delta_{i,t}^v z_{i,t+1}, \qquad (8a)$$
$$\tilde{\omega}_{i,t+1} = \omega_{i,t} + \alpha_{\omega,t} \delta_{i,t}^\omega \psi(s_t, a_t), \qquad (8b)$$

where $\alpha_{\cdot,t}$ are step sizes. Similarly, the actor parameters are updated via $\tilde{\theta}_{i,t+1}$ using the estimated advantage.

Standard methods fail as Byzantine agents (i) manipulate consensus to prevent learning and (ii) destabilize the system causing divergence. To address these, our algorithm incorporates resilient W-MSR aggregation, and synchronized stability-constrained policy mixture as follows.

First, we use **resilient W-MSR aggregation** to secure the global estimation. To enforce consistency, agents must fuse their intermediate estimates $\{\tilde{v}_{i,t+1}, \tilde{\omega}_{i,t+1}\}$. However, standard weighted averaging is vulnerable to Byzantine sabotage. Following the resilience principles for networked agents, we employ a coordinate-wise Weighted Mean-Subsequence-Reduced (W-MSR) aggregation rule. For each $k$, agent $i$ collects the set of values $\mathcal{V}_i^{(k)} = \{\tilde{v}_{j,t+1}^{(k)}\}_{j \in \mathcal{N}_i \cup \{i\}}$ and sorts them. Assuming an upper bound

$H$ on the number of Byzantine neighbors, agent $i$ removes the $H$ largest and $H$ smallest values from $\mathcal{V}_i^{(k)}$, except for those that are smaller (resp. larger) than its own value $\tilde{v}_{i,t+1}^{(k)}$. The updated parameter $v_{i,t+1}$ is computed as the average of the remaining values $\bar{\mathcal{V}}_i^{(k)}$:

$$
v_{i,t+1}^{(k)} = \frac{\sum_{j \in \bar{\mathcal{V}}_i^{(k)}} c_t(i,j) \tilde{v}_{j,t+1}^{(k)}}{\sum_{j \in \bar{\mathcal{V}}_i^{(k)}} c_t(i,j)}. \tag{9}
$$

Analogously, we define $\Omega_i^{(k)}$ and $\bar{\Omega}_i^{(k)}$ and let

$$
\omega_{i,t+1}^{(k)} = \frac{\sum_{j \in \bar{\Omega}_i^{(k)}} c_t(i,j) \tilde{\omega}_{j,t+1}^{(k)}}{\sum_{j \in \bar{\Omega}_i^{(k)}} c_t(i,j)}. \tag{10}
$$

And this mechanism ensures resilience provided the network connectivity is $(2H + 1)$-robust, as Byzantine agents can no longer arbitrarily skew the aggregate beyond the range of honest agents.

Second, we adopt **synchronized stability-constrained policy mixture** to ensure geometric ergodicity. To prevent both the queuing process and the learning process from diverging, we need to guarantee that the MMDP is ergodic. To this end, we say that a policy $\pi_i^{\text{safe}}$ is *safe* if the MMDP is geometrically ergodic under it, i.e., if there exists a probability measure $\mu$ on $\mathcal{S}$ and constants such that for every initial condition $s \in \mathcal{S}$, the total variation distance $\|P_{\pi^{\text{safe}}}^t(s, \cdot) - \mu(\cdot)\|_{\text{TV}}$ decays geometrically as $t \to \infty$. Standard independent mixing (where each agent flips a private coin) fails in decentralized MARL because the resulting joint transition kernel becomes a product measure. To address this, we introduce a synchronized mixture mechanism using a shared randomness source $b_t$ (e.g., a common seed). Agents execute a mixture of their learning policy $\pi_i$ and a known safe baseline policy $\pi_i^{\text{safe}}$:

$$
\begin{aligned}
\pi_{i,\theta}^{\text{mix}}(\cdot \mid s_t) = &b_t \, \pi_i(\cdot \mid s_t; \theta_{i,t}) \\
&+ (1 - b_t) \, \pi_i^{\text{safe}}(\cdot \mid s_t),
\end{aligned} \tag{11}
$$

where $b_t \sim \text{Bernoulli}(1/\|s\|)$. A key step to implement the above is to find a safe policy or to determine whether a policy is safe; see Proposition 3.9 in the next section.

## 3. Stability Analysis

### 3.1. Assumptions

We first assume that the service system can be stabilized; otherwise the traffic state and the learning algorithm would be unstable even without Byzantine agents. For service systems with affinities, stabilizability is not as straightforward as in the classical settings (Foley & McDonald, 2001; Kleinrock & Levy, 1988). To this end, we assume a sufficient condition for stabilizability:

**Assumption 3.1.** There exist a policy profile $\pi^{\text{safe}}$, known to the cooperative agents, such that there exists a measurable set $C \in \mathcal{S}$, a constant $t_C \in \mathbb{Z}_{\geq 0}$, and a non-trivial measure $\nu_C$ on $\mathcal{S}$ such that for all $s \in C$ and for all measurable set $C' \in \mathcal{S}$, $P_{\pi^{\text{safe}}}^m(s, B) \geq \nu_C(B)$. Furthermore, there exist constants $\{\beta_i > 0; i \in I\}$, $\rho \in (0, 1)$ and $B < \infty$ such that for every $s = (x, f) \in \mathcal{S}$,

$$
P_{\pi^{\text{safe}}} \left( \sum_{i \in \mathcal{N}} e^{\beta_i |x_i|} \right) \leq \rho \sum_{i \in \mathcal{N}} e^{\beta_i |x_i|} + B.
$$

The above conditions ensure that the service system is geometrically ergodic under $\pi^{\text{safe}}$ (see Proposition 3.9), called the *safe policy profile*. In practice, $\pi^{\text{safe}}$ can be a set of simple rules such as the join-the-shortest-queue for routers (Foley & McDonald, 2001) and exhaustive service for servers (Kleinrock & Levy, 1988). To ensure exploration, softmax versions of the above rules can be used. The function $\sum_{i \in \mathcal{N}} e^{\beta_i |x_i|}$ can be interpreted as a Lyapunov function for drift analysis.

Accordingly, throughout the stability analysis, any action execution guided by $\theta$ is assumed to follow the mixture policy $\pi_\theta^{\text{mix}}$ in (11). The following assumptions specify the regularities of several functions involved in the algorithm.

**Assumption 3.2.** The reward $r(s, a)$ is globally Lipschitz.

**Assumption 3.3.** The policy $\theta \mapsto \pi_\theta^{\text{mix}}(a|s)$ is twice differentiable and globally Lipschitz.

**Assumption 3.4.** Let $\phi$ and $\psi$ be the vectors of basis functions for value and reward approximation, respectively. There exists a polynomial $\mathcal{P}(\|s\|)$ such that for all $s \in \mathcal{S}$ and all $a \in \mathcal{A}$, $\|\phi(s,a)\| \leq \mathcal{P}(\|s\|)$ and $\|\psi(s,a)\| \leq \mathcal{P}(\|s\|)$. Furthermore, for all $a \in \mathcal{A}$, $\lim_{\|s\| \to \infty} \|\phi(s,a)\| = \infty$ and $\lim_{\|s\| \to \infty} \|\psi(s,a)\| = \infty$.

Importantly, we allow unbounded reward and basis functions, which are natural choices for service systems; this is different from the standard assumption of boundedness (Konda & Tsitsiklis, 1999). Assumption 3.4 essentially ensures that the basis functions are upper-bounded by the Lyapunov function in Assumption 3.1. The radially unboundedness ensures the non-singularity of critic updates, A key consequence of this assumption that our policy mixture mechanism in (11) will structurally enforce geometric ergodicity (Proposition 3.9).

The next assumptions are standard for decentralized actor-critic methods (Konda & Tsitsiklis, 1999; Borkar, 2008; Zhang et al., 2018):

**Assumption 3.5.** The step-size sequences $\{\alpha_{x,t}\}_{t \geq 0}$ for $x \in \{v, \omega, \theta\}$ are positive and satisfy

$$
\sum_t \alpha_{x,t} = \infty, \quad \sum_t \alpha_{x,t}^2 < \infty, \quad \lim_{t \to \infty} \frac{\alpha_{x,t+1}}{\alpha_{x,t}} = 1.
$$

Furthermore, $\alpha_{\theta,t} = o(\alpha_{v,t} + \alpha_{\omega,t})$, and $\sup_t(\alpha_{v,t}/\alpha_{\omega,t} + \alpha_{\omega,t}/\alpha_{v,t}) < \infty$.

**Assumption 3.6.** For each cooperative agent $i \in \mathcal{N}^+$, the actor update involves a projection $\Psi_{\Theta^i}$ onto a compact and hyperrectangular set $\Theta^i \subset \mathbb{R}^{m_i}$.

The last two assumptions are standard for Byzantine agents (LeBlanc et al., 2013; Ye et al., 2024):

**Assumption 3.7.** The network topologies $\mathcal{G}$ and the corresponding time-varying effective consensus matrices $\{C_t\}_{t\geq 0} \in \mathbb{R}^{N \times N}$ induced by the resilient aggregation satisfy: (i) $\mathcal{G}$ comprises a maximum of $H$ Byzantine agents; (ii) the graph $\mathcal{G}$ is $(2H+1)$-robust; (iii) $C_t$ is consistent with the trimmed topology $\mathcal{G}'_t$: $c_t(i,j) \geq \nu > 0$ if $(i,j) \in \mathcal{G}'_t$, and $0$ otherwise; (iv) he matrix $C_t$ is row-stochastic, i.e., $C_t \mathbf{1} = \mathbf{1}$.

**Assumption 3.8.** For every Byzantine agent $i \in \mathcal{N}^-$, the policy sequence converges to a stationary strategy $\pi_*^k$ with rate $|\pi_{t+1}^i - \pi_t^i| = O(\alpha_{\theta,t})$.

### 3.2. Main results

To state the main results, we first define some notations to make the presentation more compact and intuitive. Let $\mathbf{w}_{i,t} \triangleq [v_{i,t}^\top, \omega_{i,t}^\top]^\top \in \mathbb{R}^{d_v + d_\omega}$. Let $\phi_t \triangleq \phi(s_t)$ and $\psi_t \triangleq \psi(s_t, a_t)$ and

$$
G_{\theta_t} \triangleq \begin{bmatrix} -z_{i,t+1}(\gamma P_\pi \phi_t - \phi_t)^\top & 0 \\ 0 & \frac{\alpha_{\omega,t}}{\alpha_{v,t}} \psi_t \psi_t^\top \end{bmatrix},
$$
$$
h_{\theta_t}^i \triangleq \begin{bmatrix} z_{i,t+1} r_{i,t+1} \\ \frac{\alpha_{\omega,t}}{\alpha_{v,t}} \psi_t \, r_{i,t+1} \end{bmatrix}. \tag{12}
$$

Thus, the updates (8a)–(8b) can be compactly written as

$$
\tilde{\mathbf{w}}_{i,t+1} = \mathbf{w}_{i,t} + \alpha_{v,t}\left( h_{\theta_t}^i - G_{\theta_t}\mathbf{w}_{i,t} + \xi_{i,t+1}\mathbf{w}_{i,t} \right), \tag{13}
$$

where $\xi_{i,t+1} \triangleq \mathrm{diag}(\gamma z_{i,t+1}(\phi_{t+1} - P_\pi \phi_t)^\top, 0)$. By Assumption 3.8, the update of cooperative agents can be characterized by a row-stochastic matrix $C_t^+ \in \mathbb{R}^{N^+ \times N^+}$ restricted to the cooperative subgraph; see Proposition B.3. We define the averaging matrix $J \triangleq \frac{1}{N^+}\mathbf{1}_{N^+}\mathbf{1}_{N^+}^\top$ and the disagreement projection $J_\perp \triangleq I_{N^+} - J$.

Our first result states that the synchronized policy mixture ensures safety:

**Proposition 3.9.** *Suppose that Assumptions 3.1 hold. Under the synchronized mixture (11), the MMDP remains geometrically ergodic for any learning policy $\pi^\theta$, ensuring bounded state moments for learning.*

A consequence of this proposition is the existence of a unique stationary state-action distribution $\eta_\theta$ associated with a given $\theta$. Thus, we can define $G(\theta) \triangleq \mathrm{E}_{\eta_\theta}[G_\theta]$, $h(\theta) \triangleq$

$\mathrm{E}_{\eta_\theta}[[(h_\theta^1)^\top, \cdots, (h_\theta^{N^+})^\top]^\top]$ and

$$
g_t^i(\theta) = \mathrm{E}_{(s,a,s') \sim \eta_\theta, P}\Big( (\psi(s,a)^\top \omega_t^i + \gamma\phi(s')^\top v_t^i
$$
$$
- \phi(s)^\top v_t^i) \cdot \nabla_{\theta^i} \log \pi^i(a^i \mid s; \theta^i) \Big) \tag{14}
$$
$$
\bar{g}^i(\theta) = \left\{ \lim_{n,m\to\infty} \frac{1}{m} \sum_{t=n}^{n+m-1} g_t^i(\theta) \right\}.
$$

Our second result establishes the convergence of the critic weights:

**Theorem 3.10.** *Suppose that Assumptions 3.1–3.8 hold. Let $\mathbf{w}^*(\theta)$ denote the unique solution to the mean-field linear system $G(\theta)\mathbf{w} = (\frac{1}{N^+}\mathbf{1}_{N^+}^\top \otimes I_d)h(\theta)$. Then, for every cooperative agent $i \in \mathcal{N}^+$, there exists a constant $\delta > 0$ such that*

$$
\limsup_{t\to\infty} \|\mathbf{w}_{i,t} - \mathbf{w}^*(\theta_t)\|_\infty
$$
$$
\leq \frac{2}{\delta N^+} \limsup_{t\to\infty} \Big\| \big(\mathbf{1}^\top(C_t^+ - J) \otimes I_d\big)h(\theta_t)
$$
$$
+ \big(\mathbf{1}^\top C_t^+ \otimes I_d\big)(J_\perp \otimes I_d)\frac{\mathbf{w}_t}{\alpha_{v,t}} \Big\|_\infty \quad a.s. \tag{15}
$$

*In particular, the right-hand side of (15) vanishes as $H \to 0$.*

Theorem 3.10 establishes that in the presence of Byzantine agents, the critic parameters converge to a bounded neighborhood of the mean linear system's solution, rather than the exact equilibrium. The radius of this neighborhood is primarily governed by the magnitude of the adversarial perturbations.

Our next result establishes the convergence of the actor parameters:

**Theorem 3.11.** *Suppose that Assumptions 3.1–3.8 hold. Then the sequence of joint policy parameters $\{\boldsymbol{\theta}_t\}$ converges with probability one to the limit set of the coupled projected differential inclusion:*

$$
\dot{\theta}^i(t) \in \Psi_{\Theta^i}\Big[\bar{g}^i(\boldsymbol{\theta}(t))\Big], \quad \forall i \in \mathcal{N}^+. \tag{16}
$$

Theorem 3.11 establishes convergence to an invariant set rather than a single point, with a diameter determined by the critic's residual estimation bias. In the absence of attacks, this residual bias vanishes, causing the limit set to collapse to the stationary points of $J^+$, thereby recovering classical convergence results.

The proofs of the above results are provided in Appendices B–C.

# 4. Case Studies

## 4.1. Scenarios and setup

We evaluate the proposed resilient decentralized MARL framework on three representative service systems:

*(i)* Large language model semantic routing. This scenario models a decentralized inference service with multiple routers and parallel GPU servers. Routers independently route heterogeneous semantic requests to downstream servers with FIFO queues. A key feature is explicit *task–server affinity*: KV-cache reuse induces higher service rates when requests are consistently routed to compatible GPUs. Preserving this affinity is critical for specialization and queue stability, while adversarial disruptions that corrupt affinity rapidly propagate into system-wide instability.

*(ii)* Multi-agent edge computing. We consider a distributed polling system where multiple edge servers autonomously select polling policies to serve geographically distributed task sources under stochastic arrivals. Affinity emerges implicitly as persistent *source–server alignment*, enabling servers to reduce switching overhead and stabilize service capacity. Byzantine perturbations repeatedly break this alignment, leading to inefficient switching behavior and accelerated queue growth.

*(iii)*Smart manufacturing delivery. This setting models a grid-based delivery system in which mobile agents transport tasks under battery and charging constraints. Affinity arises as spatial and operational specialization between agents and demand queues. When such specialization is disrupted by adversarial manipulation or unstable learning, agents incur inefficient routes and energy usage, amplifying congestion and undermining stability.

Across all systems, we evaluate learning stability, service performance, and affinity under nominal operation, Byzantine attacks, and the proposed defense (i.e., policy mixture). All experiments are averaged over 10 independent runs. And each experiment is run for $T$ episodes with 100 steps per episode. Byzantine agents constitute $20\%$ of the population unless otherwise stated, and can arbitrarily manipulate communicated value or policy signals.

We compare three settings: (i) no Byzantine agents, (ii) Byzantine agents without policy mixture, and (iii) Byzantine attack with policy mixture. All hyperparameters are kept identical across settings to isolate the effect of adversarial interference and defense.

## 4.2. Stability Analysis

Figure 3 illustrates the evolution of learning-related quantities under the three settings. In the nominal case, learning dynamics are well behaved, with temporal-difference errors and parameter updates converging smoothly. In contrast, the presence of Byzantine agents induces persistent oscillations or divergence in learning signals, indicating that malicious information exchange corrupts decentralized value estimation. When the proposed resilient algorithm is enabled, learning stability is largely restored: fluctuations are significantly damped and convergence behavior comparable to the nominal case is recovered. These results confirm that Byzantine perturbations primarily destabilize the learning process through the information layer, and that parameter-level consensus combined with policy mixture is effective in suppressing this effect.

The impact of learning instability on physical service dynamics is shown in Figure 4, which reports the evolution of average queue lengths. Without Byzantine agents, queues remain bounded and fluctuate around a stable operating point. Under Byzantine attack without defense, queue lengths grow rapidly and eventually diverge, reflecting a breakdown of effective control. This divergence highlights a critical coupling between corrupted learning updates and unbounded physical state evolution. With the proposed defense, queue growth is effectively contained and the system operates in a stable regime, despite the continued presence of Byzantine agents. This demonstrates that stabilizing learning updates is sufficient to prevent state drift to infinity in unbounded service systems.

*Table 1.* Average service time per task across three cases.

| Scenario | No attack | Attack only | Attack+defense |
|----------|-----------|-------------|----------------|
| LLM | 6.59 | 4.46e+5 | 6.88 |
| MEC | 3.60 | 18.16 | 4.13 |
| Delivery | 4.37 | 36.28 | 4.48 |

The stabilization effects observed above translate directly into improved service performance. Table 1 summarizes a key metric, the end-to-end service time per task, across the three settings. While Byzantine attacks severely degrade performance in the undefended case, the proposed algorithm recovers a substantial fraction of nominal performance. Although a performance gap relative to the no-attack baseline remains, the results indicate that preventing instability and state explosion is sufficient to maintain practically acceptable service quality.

## 4.3. Affinity analysis.

Table 2 summarizes the average learned task–server affinity across three representative service systems under nominal operation, Byzantine attack, and attack with defense. In the absence of Byzantine agents, all systems exhibit consistently high affinity values, indicating that decentralized MARL successfully captures long-term compatibility between task types and service nodes. This emergent affinity structure explains the stable learning behavior and bounded queue dynamics observed in the nominal experiments, as affinity-

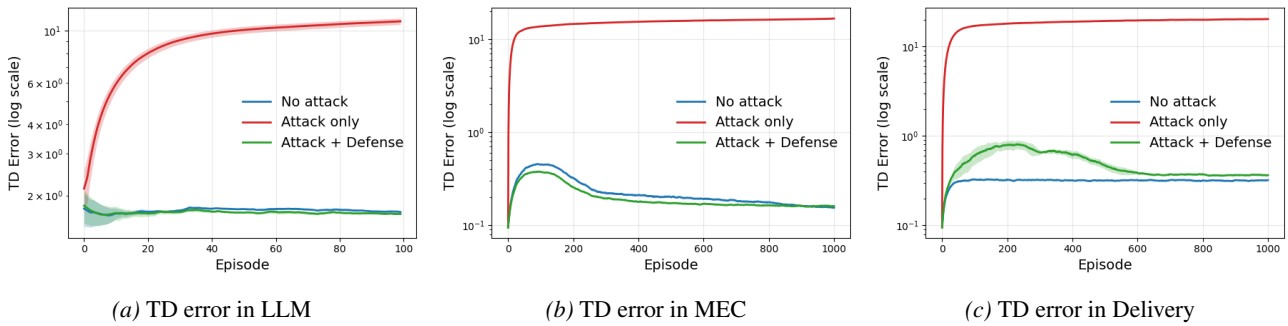

*(a) TD error in LLM*       *(b) TD error in MEC*       *(c) TD error in Delivery*

*Figure 3.* Temporal-difference (TD) error during training under Byzantine attacks. Across all three systems, Byzantine agents induce persistent TD error growth, indicating unstable learning. The proposed defense suppresses error amplification and restores stable convergence, comparable to the no-attack case.

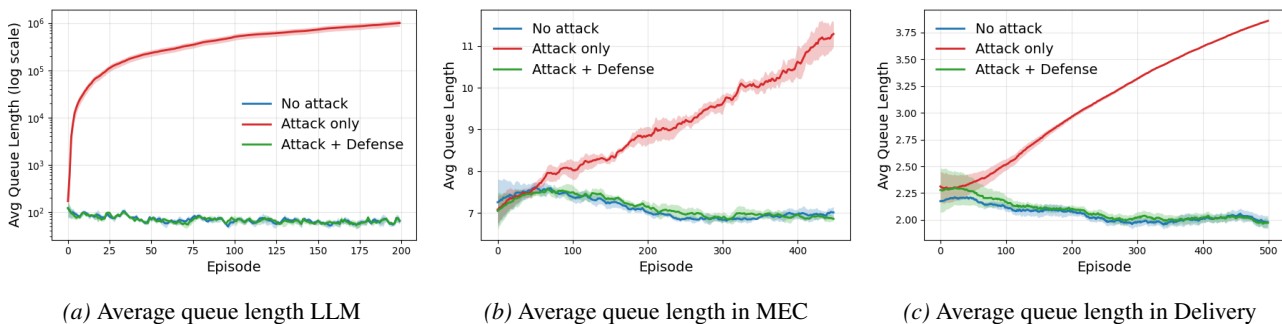

*(a) Average queue length LLM*     *(b) Average queue length in MEC*     *(c) Average queue length in Delivery*

*Figure 4.* Average queue length during training under Byzantine attacks. Across all systems, Byzantine agents cause queue divergence without defense, while the proposed defense preserves queue stability and attains average queue lengths close to the no-attack cases.

aligned decisions promote specialization, reduce contention, and improve overall service efficiency.

*Table 2.* Average learned task–server affinity across three cases.

| Scenario | No attack | Attack only | Attack+defense |
|----------|-----------|-------------|----------------|
| LLM | 0.801 | 0.472 | 0.664 |
| MEC | 0.785 | 0.351 | 0.642 |
| Delivery | 0.633 | 0.409 | 0.580 |

When Byzantine agents are introduced without defense, affinity deteriorates sharply across all scenarios. Rather than merely increasing variance, adversarial information exchange disrupts the learning of persistent task–server compatibility, leading to flattened or inconsistent affinity patterns. This structural degradation provides a unifying explanation for the learning instability and queue divergence observed earlier: once affinity is corrupted, routing and scheduling decisions lose their long-term coherence, amplifying congestion and inducing unbounded state drift.

### 4.4. Ablation Study

To better understand the contribution of each component, we evaluate two ablation variants: (i) W-MSR without synchronized policy mixture, and (ii) synchronized policy mixture without W-MSR.

Figure 5 shows the corresponding queue dynamics and learning behavior across the three scenarios. Using only W-MSR improves robustness against adversarial parameter corruption, but unstable queue growth still appears in highly dynamic environments due to uncontrolled state evolution. In contrast, policy mixture alone preserves bounded queue behavior by stabilizing the physical system dynamics, but learning quality degrades substantially because corrupted parameter updates are not filtered.

The full method consistently achieves the best overall performance, combining stable queue dynamics with robust decentralized learning. These results confirm that the two components address complementary failure modes: W-MSR protects the information layer from Byzantine corruption, while synchronized policy mixture stabilizes the underlying queueing dynamics.

### 4.5. Comparison with Robust Aggregation Baselines

We further compare the proposed method against several representative robust aggregation methods, including mean aggregation, coordinate-wise median aggregation, and projection-based aggregation.

Figure 5 summarizes the performance comparison across the three scenarios. In the MEC scenario, the proposed

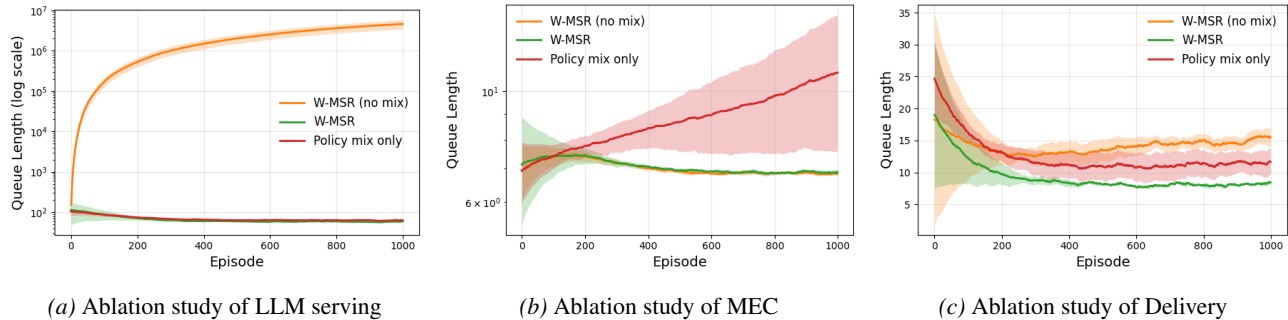

*(a)* Ablation study of LLM serving      *(b)* Ablation study of MEC      *(c)* Ablation study of Delivery

*Figure 5.* Ablation study under Byzantine attacks across the three service systems. Using W-MSR without policy mixture improves robustness against adversarial parameter corruption but cannot fully prevent unstable queue growth. Using only synchronized policy mixture preserves bounded queue behavior but suffers from degraded learning due to corrupted updates. The full method consistently achieves the best balance between learning stability and queue stability, demonstrating that the two components address complementary failure modes.

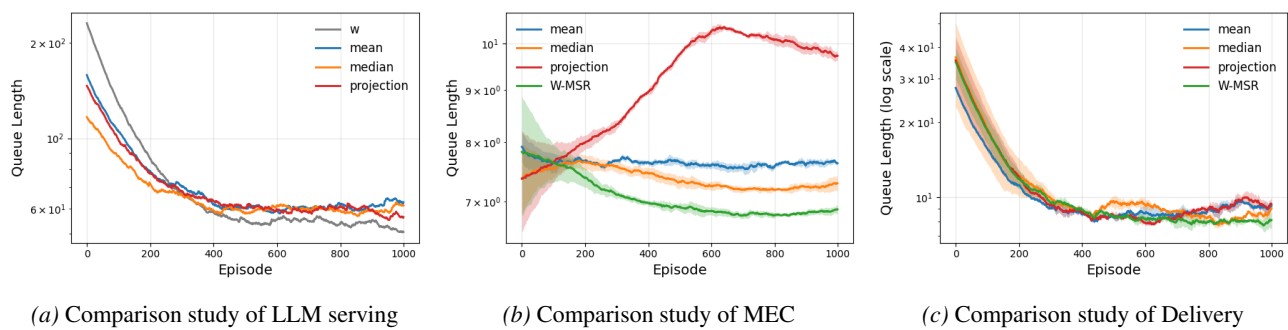

*(a)* Comparison study of LLM serving      *(b)* Comparison study of MEC      *(c)* Comparison study of Delivery

*Figure 6.* Comparison with representative robust aggregation baselines under Byzantine attacks. The proposed method consistently achieves strong performance across all scenarios, particularly in highly dynamic environments where queue stability and learning robustness are tightly coupled. Compared with existing robust aggregation methods, the proposed framework better balances adversarial robustness and stable queue evolution.

method significantly outperforms all baselines under Byzantine attacks, demonstrating strong robustness in highly dynamic environments. In the LLM routing scenario, our method achieves performance comparable to existing robust methods, with slightly more conservative behavior due to stronger stabilization effects. In the delivery scenario, all methods exhibit similar performance, suggesting that this environment is inherently less sensitive to adversarial perturbations.

Overall, these results indicate that the proposed framework is particularly advantageous in adversarial-sensitive systems with strong coupling between learning dynamics and queue stability, while remaining competitive in more stable environments.

## 5. Conclusions

This paper studied decentralized MARL for service systems with job–server affinity under Byzantine attack and unbounded spaces. By integrating Byzantine resilient parameter consensus with a synchronized stability-constrained policy mixture, we achieved stable queuing and robust learning dynamics with provable convergence guarantees. Case studies in LLM semantic routing, multi-agent edge computing, and smart manufacturing demonstrate the generality and practical relevance of the proposed framework. Future work includes extending the algorithm to handle stochastic perturbations or adaptive adversaries, and robustness to model and communication uncertainties.

## Impact Statement

This paper presents work whose goal is to advance the field of Machine Learning. There are many potential societal consequences of our work, none which we feel must be specifically highlighted here.

## Acknowledgements

This work was in part supported by the National Natural Science Foundation of China under Grant 62473250.

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

# Appendices

## Contents

## A. Pseudocode of Proposed Algorithm

---

**Algorithm 1** Resilient Decentralized Actor-Critic with W-MSR and Stability Mixture

---

1: **Input:** Graph $\mathcal{G} = (\mathcal{N}, \mathcal{E})$, parameter $H$ (Byzantine bound), safe policy $\pi^{\text{safe}}$, step sizes $\alpha_{\theta,t}, \alpha_{v,t}, \alpha_{\omega,t}$.
2: **Initialize:** Parameters $\theta_i, v_i, \omega_i$ and eligibility trace $z_i = 0$ for all agents $i \in \mathcal{N}$. Shared randomness source (seed).
3: **for** time step $t = 0, 1, 2, \ldots$ **do**
4:     Observe global state $s_t$.
5:     **for** each cooperative agent $i \in \mathcal{N}^+$ **do**
6:         // **1. Synchronized Policy Mixture (Eq. 11)**
7:         Sample shared coordination variable $b_t \sim \text{Bernoulli}(\min(1, 1/\|s_t\|))$.
8:         **if** $b_t = 1$ **then**
9:             Sample action $a_{i,t} \sim \pi_i(\cdot|s_t; \theta_{i,t})$ (Exploration/Learning).
10:         **else**
11:             Sample action $a_{i,t} \sim \pi_i^{\text{safe}}(\cdot|s_t)$ (Stability enforcement).
12:         **end if**
13:     **end for**
14:     Execute joint action $\mathbf{a}_t$, observe next state $s_{t+1}$ and local rewards $\{r_{i,t}\}$.
15:     **for** each cooperative agent $i \in \mathcal{N}^+$ **do**
16:         // **2. Local Critic Update & Eligibility Traces**
17:         Compute TD errors $\delta_{i,t}^v$ and $\delta_{i,t}^\omega$ via Eq. (7).
18:         Update trace: $z_{i,t+1} = \gamma\lambda z_{i,t} + \phi(s_t)$.
19:         Compute **intermediate estimates** via Eq. (8a)-(8b):
20:             $\tilde{v}_{i,t+1} \leftarrow v_{i,t} + \alpha_{v,t}\delta_{i,t}^v z_{i,t+1}$
21:             $\tilde{\omega}_{i,t+1} \leftarrow \omega_{i,t} + \alpha_{\omega,t}\delta_{i,t}^\omega \psi(s_t, a_t)$
22:         // **3. Resilient W-MSR Aggregation**
23:         Send $\{\tilde{v}_{i,t+1}, \tilde{\omega}_{i,t+1}\}$ to neighbors $j \in \mathcal{N}_i$; Receive neighbors' estimates.
24:         **for** each coordinate $k$ **do**
25:             Collect $\mathcal{V}_i^{(k)} = \{\tilde{v}_{j,t+1}^{(k)}\}_{j\in\mathcal{N}_i\cup\{i\}}$.
26:             Sort $\mathcal{V}_i^{(k)}$ and remove the $H$ largest and $H$ smallest values (Coordinate-wise W-MSR).
27:             Update $v_{i,t+1}^{(k)}$ as the weighted average of remaining values Eq.(9).
28:             Perform analogous W-MSR aggregation for $\omega_{i,t+1}^{(k)}$ Eq.(10).
29:         **end for**
30:         // **4. Actor Parameter Update**
31:         Estimate global advantage $A^\pi(s_t, a_t)$ using aggregated $v_{i,t+1}$ and $\omega_{i,t+1}$.
32:         Update $\theta_{i,t+1} \leftarrow \theta_{i,t} + \alpha_{\theta,t}\nabla_{\theta_i}\log\pi_i(a_{i,t}|s_t; \theta_{i,t})A^\pi(s_t, a_t)$.
33:     **end for**
34: **end for**

---

## B. Technical Conditions for Stochastic Approximation

In this appendix, we establish the rigorous mathematical framework required for the convergence analysis of our algorithm. The content is organized as follows:

- In Appendix B.1, we verify the technical conditions related to the critic update, including the regularity of the MDP, the stability of the critic system, and the contraction properties of the resilient W-MSR consensus mechanism.

- In Appendix B.2, we state the general convergence theorem for constrained stochastic approximation, which serves as the theoretical basis for the actor convergence proof.

### B.1. Properties of the Critic Update and Consensus Mechanism

In this subsection, we formally state the technical assumptions and lemmas required to analyze the critic updates. We cast the critic update into a general linear stochastic approximation framework driven by a Markov chain, following the

formalism of Konda & Tsitsiklis (2003b).

Consider the joint state-action-consensus process $\mathbf{Y}_t$ evolving on a Polish space $\mathcal{Y}$. The critic parameter $\mathbf{w}_t \in \mathbb{R}^{d_{crit}}$ and actor parameter $\theta_t \in \mathbb{R}^{d_{act}}$ are updated according to:

$$\tilde{\mathbf{w}}_{t+1}^i = \mathbf{w}_t^i + \alpha_{v,t} \left( h_{\theta_t}(\mathbf{Y}_{i,t+1}) - G_{\theta_t}(\mathbf{Y}_{i,t+1})\mathbf{w}_t + \xi_{i,t+1} \right), \tag{17}$$
$$\theta_{t+1} = \theta_t + \alpha_{\theta,t} H_{t+1}, \tag{18}$$

**Regularity of the MDP and Poisson Solutions**   First, we establish the existence of solutions to the Poisson equation and the boundedness of moments. These are direct consequences of the uniform geometric drift condition.

**Lemma B.1** (Regularity of Update Fields). *Consider the joint state-action-consensus process $\mathbf{Y}_t$. Under Assumption 3.3 (Uniform Geometric Ergodicity) and Assumption 3.4 (Polynomial Growth), the following properties hold for any $\theta \in \Theta$:*

(a) ***Existence of Poisson Solutions:*** *There exist deterministic mean fields $\bar{h}(\theta), \bar{G}(\theta)$ and measurable functions $\hat{h}_{\theta}(\cdot), \hat{G}_{\theta}(\cdot)$ solving the Poisson equations defined in Assumption A.1 of Konda & Tsitsiklis (2003b):*

$$\hat{h}_{\theta}(y) = h_{\theta}(y) - \bar{h}(\theta) + (P_{\theta}\hat{h}_{\theta})(y),$$
$$\hat{G}_{\theta}(y) = G_{\theta}(y) - \bar{G}(\theta) + (P_{\theta}\hat{G}_{\theta})(y).$$

(b) ***Uniform Moment Bounds:*** *The update functions $h, G$ and their Poisson solutions $\hat{h}, \hat{G}$ belong to the function class $\mathcal{D}$ uniformly. Specifically, for any $q \geq 1$, there exists $D_q < \infty$ such that:*

$$\sup_{t \geq 0} \mathbb{E}\left[\|\hat{h}_{\theta_t}(\mathbf{Y}_t)\|^q + \|\hat{G}_{\theta_t}(\mathbf{Y}_t)\|^q\right] \leq D_q.$$

(c) ***Lipschitz Continuity:*** *The mean fields $\bar{h}(\theta)$ and $\bar{G}(\theta)$ are Lipschitz continuous in $\theta$. Furthermore, the transition kernel satisfies the Lipschitz condition on Poisson solutions:*

$$\|(P_{\theta}\hat{f}_{\theta})(y) - (P_{\theta'}\hat{f}_{\theta'})(y)\| \leq K(y)\|\theta - \theta'\|,$$

*where $K(y) \in \mathcal{D}$ and $\hat{f} \in \{\hat{h}, \hat{G}\}$.*

*Proof.* This result follows directly from Lemma 4.3 and Proposition 4.4 in Konda & Tsitsiklis (2003b). The geometric drift (Assumption 3.3) ensures the existence of the Poisson solutions, while the differentiability of the policy and features (Assumptions 3.3 and 3.4) ensures Lipschitz continuity. □

**Stability of the Critic System**   The convergence of the linear critic iteration relies on the mean-field matrix being Hurwitz (stable). We explicitly derive this from the feature rank assumption, which follows directly from Lemma 5.6 in Konda & Tsitsiklis (2003b).

**Lemma B.2** (Uniform Positive Definiteness). *There exists a constant $\delta > 0$ such that for all $\theta \in \Theta$ and all vectors $u \in \mathbb{R}^{N^+ d}$:*

$$u^\top G(\theta)u \geq \delta\|u\|^2. \tag{19}$$

**Properties of Resilient Consensus**   Finally, we characterize the properties of the W-MSR mechanism. We formally state that the non-linear W-MSR update on a robust graph is equivalent to a linear matrix update that induces contraction.

**Proposition B.3** (Equivalent Matrix Representation and Contraction). *Under Assumption 3.7 ($(2H+1)$-robustness), the resilient consensus update for cooperative agents satisfies the following properties:*

1. ***Matrix Representation:*** *For each time $t$, there exists a row-stochastic matrix $C_t^+ \in \mathbb{R}^{N^+ \times N^+}$ such that the update of the cooperative agents is mathematically equivalent to:*

$$\mathbf{w}_{t+1}^+ = (C_t^+ \otimes I_d)\tilde{\mathbf{w}}_t^+.$$

2. **Significant Weights:** *There exists a scalar $\nu > 0$ such that for all $i, j \in \mathcal{N}^+$, if agent $j$ is included in agent $i$'s update (including self-loops $i = j$), then:*

$$[C_t^+]_{ij} \geq \nu/2 \quad and \quad [C_t^+]_{ii} \geq \nu.$$

3. **Contraction in Disagreement Space:** *Let $J \triangleq \frac{1}{N^+} \mathbf{1}\mathbf{1}^\top$ and $J_\perp \triangleq I_{N^+} - J$. The sequence $\{C_t^+\}$ induces a mean contraction in the error subspace. Specifically, there exists $\rho \in [0, 1)$ such that:*

$$\sup_{t \geq 0} \lambda_{\max} \left( \mathbb{E} \left[ (C_t^+)^\top J_\perp C_t^+ \mid \mathcal{F}_t \right] \right) \leq \rho. \tag{20}$$

*Proof.* Items 1 and 2 are established in Sundaram & Gharesifard (2018, Proposition 5.1) as a consequence of the W-MSR pruning mechanism on a robust graph. Item 3 is established in Ye et al. (2024, Lemma 23) and follows from the connectivity of the cooperative subgraph (implied by robustness) and the uniform lower bound on weights, which ensures that the product of these stochastic matrices contracts towards the consensus subspace geometrically in expectation. $\square$

**Martingale Noise Properties** Recall that the global critic update is driven by the aggregated noise vector $\boldsymbol{\xi}_{t+1}$. We formally define this vector as the concatenation of the local noise terms from all cooperative agents:

$$\boldsymbol{\xi}_{t+1} \triangleq \left[ (\xi_{1,t+1} \mathbf{w}_{1,t})^\top, \ldots, (\xi_{N^+,t+1} \mathbf{w}_{N^+,t})^\top \right]^\top. \tag{21}$$

**Lemma B.4** (Martingale Difference). *The noise sequence $\{\boldsymbol{\xi}_{t+1}\}$ defined in the critic update is a martingale difference sequence with respect to $\mathcal{F}_t$, satisfying $\sup_t \mathbb{E}[\|\boldsymbol{\xi}_{t+1}\|^2 \mid \mathcal{F}_t] < \infty$ almost surely.*

*Proof.* The zero-mean property follows by definition (sample update minus conditional expectation). The moment boundedness follows from Lemma B.1(b), as $\boldsymbol{\xi}_{t+1}$ is a linear combination of functions in class $\mathcal{D}$. $\square$

### B.2. General Theorem for Constrained Stochastic Approximation

For completeness, we state the general convergence theorem for constrained stochastic approximation algorithms driven by martingale difference noise and Markovian bias. This framework is adapted from Kushner & Yin (2003, Chapter 5), generalizing the standard ODE method to differential inclusions.

Consider the recursive update for a parameter $\theta_t \in \mathbb{R}^d$ confined to a compact hyperrectangular set $\Theta$:

$$\theta_{t+1} = \Psi_\Theta \left( \theta_t + \alpha_t \left[ g_t(\theta_t) + \delta M_{t+1} + \beta_t \right] \right), \tag{22}$$

where $\Psi_\Theta$ is the projection operator, $\alpha_t$ is the stepsize sequence, $g_t(\cdot)$ is the mean field function, $\delta M_{t+1}$ is a noise term, and $\beta_t$ is a bias term. Let $\mathcal{F}_t$ denote the filtration generated by the history up to time $t$.

**Theorem B.5** (Convergence to Differential Inclusion). *Suppose the following conditions hold:*

1. **Boundedness:** *The sequence $\{\theta_t\}$ is bounded almost surely. The mean field $g_t(\cdot)$ and noise terms have uniformly bounded moments.*

2. **Stepsizes:** *The stepsize sequence satisfies $\sum_t \alpha_t = \infty$, $\sum_t \alpha_t^2 < \infty$, and $\lim_{t \to \infty} \frac{\alpha_{t+1}}{\alpha_t} = 1$.*

3. **Martingale Noise:** *The sequence $\{\delta M_{t+1}\}$ is a martingale difference sequence with respect to $\mathcal{F}_t$, i.e., $\mathbb{E}[\delta M_{t+1} \mid \mathcal{F}_t] = 0$, with bounded conditional second moments: $\sup_t \mathbb{E}[\|\delta M_{t+1}\|^2 \mid \mathcal{F}_t] < \infty$ a.s.*

4. **Vanishing Bias:** *The bias term satisfies $\beta_t \to 0$ almost surely as $t \to \infty$.*

5. **Regularity and Limit Set:** *The function $g_t(\theta)$ is continuous in $\theta$ uniformly in $t$. Furthermore, define the set of accumulation points of the time-averaged mean fields as:*

$$\bar{g}(\theta) \triangleq \left\{ v \in \mathbb{R}^d \; \middle| \; \exists \{n_k, m_k\} \to \infty \; s.t. \; v = \lim_{k \to \infty} \frac{1}{m_k} \sum_{\tau = n_k}^{n_k + m_k - 1} g_\tau(\theta) \right\}.$$

*The set-valued map $G(\cdot)$ is upper semicontinuous (USC).*

*Then, the sequence $\{\theta_t\}$ converges with probability one to the limit set of the projected differential inclusion:*

$$\dot{\theta}(t) \in \Psi_\Theta \left[ \bar{g}(\theta(t)) \right]. \tag{23}$$

# C. Proofs of the Main Results

In this section, we provide detailed proofs for the main theoretical results presented in Section 4. The analysis proceeds in a hierarchical manner, reflecting the timescale separation and the interdependencies of the algorithm's components:

- First, we prove Proposition 3.9, establishing that the synchronized policy mixture mechanism guarantees the geometric ergodicity of the underlying MMDP. This ensures that the state moments remain bounded, which is a prerequisite for all subsequent learning analyses.

- Second, we prove Theorem 3.10. Utilizing the ergodicity established above and the contraction property of the resilient consensus (Proposition B.3), we show that the critic parameters converge to a bounded neighborhood of the ideal mean-field solution.

- Finally, we prove Theorem 3.11. By treating the convergent critic as a quasi-static estimator with bounded bias, we apply the general stochastic approximation framework (Theorem B.5) to show that the actor updates track a differential inclusion governed by the cooperative objective.

## C.1. Proof of Proposition 3.9

To establish the geometric ergodicity of the MMDP under the synchronized stability-constrained policy mixture, we invoke the Foster-Lyapunov drift criterion for general state space Markov chains (Meyn & Tweedie, 2012, Theorem 15.0.1).

*Proof.* Let $V$ be the candidate Lyapunov function defined in Assumption 3.1. By the assumption, the safe policy $\pi^{\text{safe}}$ ensures geometric drift:

$$P_{\pi^{\text{safe}}} V(s) \leq \rho V(s) + B, \quad \text{with } \rho \in (0, 1). \tag{24}$$

For the learning policy $\pi_{\boldsymbol{\theta}}$ (and any arbitrary behavior from Byzantine agents), we observe that in physical service systems, the rates of job arrivals and service completions are finite. Consequently, the change in queue lengths in a single time step is uniformly bounded. Thus, the one-step growth of the exponential Lyapunov function is bounded by some constant $K_{\max}$ for any policy:

$$P_{\boldsymbol{\theta}} V(s) \leq K_{\max} V(s), \quad \forall s \in \mathcal{S}. \tag{25}$$

Under the *synchronized* mixture mechanism defined in Eq. (11), the joint action profile is determined by a single shared Bernoulli variable $\zeta$. This implies that with probability $\epsilon(s)$, *all* agents execute the learning policy, and with probability $1 - \epsilon(s)$, *all* agents execute the safe policy. Critically, this synchronization ensures that the effective joint transition kernel $P_{\text{mix}}$ is a linear convex combination of the component kernels:

$$P_{\text{mix}}(s, \cdot) = \epsilon(s) P_{\boldsymbol{\theta}}(s, \cdot) + (1 - \epsilon(s)) P_{\pi^{\text{safe}}}(s, \cdot). \tag{26}$$

(Note: If agents mixed independently, the joint kernel would be a product of mixtures, destroying the linearity of the drift analysis).

Applying the mixture kernel to the Lyapunov function with $\epsilon(s) = 1/\|s\|$:

$$P_{\text{mix}} V(s) = \frac{1}{\|s\|} P_{\boldsymbol{\theta}} V(s) + \left(1 - \frac{1}{\|s\|}\right) P_{\pi^{\text{safe}}} V(s)$$

$$\leq \frac{1}{\|s\|} K_{\max} V(s) + \left(1 - \frac{1}{\|s\|}\right) (\rho V(s) + B)$$

$$= \left[\rho + \frac{1}{\|s\|} (K_{\max} - \rho)\right] V(s) + B \left(1 - \frac{1}{\|s\|}\right).$$

Define the effective drift coefficient $\rho_{\text{eff}}(s) = \rho + \frac{1}{\|s\|}(K_{\max} - \rho)$. As $\|s\| \to \infty$, $\rho_{\text{eff}}(s) \to \rho < 1$. Therefore, there exists a sufficiently large radius $R$ such that for all states with $\|s\| > R$, we have $\rho_{\text{eff}}(s) < 1$. This establishes the geometric drift condition outside the compact set $\mathcal{C} = \{s : \|s\| \leq R\}$.

Inside the compact set $\mathcal{C}$, the transition kernel satisfies the minorization condition. Specifically, for $\|s\| \leq R$, the probability of executing the safe policy is bounded away from zero ($1 - \epsilon(s) \geq \delta > 0$). Since the safe policy $\pi^{\text{safe}}$ satisfies the small set condition (Assumption 3.1), the mixture kernel $P_{\text{mix}}$ inherits this property.

Since the induced Markov chain is irreducible, aperiodic, and satisfies the geometric drift condition with a petite set, it is $V$-uniformly geometrically ergodic. This implies the existence of a unique stationary distribution $\mu$ with finite moments of $V$. Since $V$ is exponential, all polynomial moments of the state (required for the function class $\mathcal{F}$) are finite. $\qquad\square$

### C.2. Proof of Theorem 3.10

C.2.1. BOUNDEDNESS OF THE CRITIC ITERATES $\{\mathbf{w}_t\}$

In this subsection, we establish that the sequence of stacked critic parameters $\mathbf{w}_t = [\mathbf{w}_{1,t}^\top, \ldots, \mathbf{w}_{N^+,t}^\top]^\top \in \mathbb{R}^{N^+(d_v+d_\omega)}$ remains bounded almost surely. This stability result is a prerequisite for the convergence rate analysis. We employ the *rescaling technique* (also known as the ODE method at infinity) for stochastic approximation with multiplicative noise (Borkar, 2008).

**Global Recursion and Block-Maximum Norm.** Recall the compact local update in Eq. (13). We lift this to the network level by stacking the vectors for all cooperative agents $i \in \mathcal{N}^+$. Define the global stacked observation vector $\mathbf{h}_{\theta_t}(\mathbf{Y}_{t+1}) \triangleq [(h_{\theta_t}^1)^\top, \ldots, (h_{\theta_t}^{N^+})^\top]^\top$ and the global block-diagonal observation matrix $\mathbf{G}_{\theta_t}(\mathbf{Y}_{t+1}) \triangleq \mathrm{blkdiag}(G_{\theta_t}, \ldots, G_{\theta_t})$. Similarly, let $\mathbf{\Xi}_{t+1} \triangleq \mathrm{blkdiag}(\xi_{1,t+1}, \ldots, \xi_{N^+,t+1})$ denote the stacked multiplicative martingale noise. The global recursion evolves as:

$$\mathbf{w}_{t+1} = (C_t^+ \otimes I_d)\left(\mathbf{w}_t + \alpha_{v,t}\left[\mathbf{h}_{\theta_t}(\mathbf{Y}_{t+1}) - \mathbf{G}_{\theta_t}(\mathbf{Y}_{t+1})\mathbf{w}_t + \mathbf{\Xi}_{t+1}\mathbf{w}_t\right]\right). \tag{27}$$

Let $h(\theta)$ and $G(\theta)$ be the mean fields defined in the main text (expectations under $\eta_\theta$). We define the corresponding global mean field matrix as $\mathbf{G}(\theta) \triangleq I_{N^+} \otimes G(\theta)$ and the global mean vector as $\mathbf{h}(\theta) \triangleq \mathbf{1}_{N^+} \otimes h(\theta)$. Note that $\mathbf{G}(\theta)$ inherits the uniform positive definiteness from $G(\theta)$ (Lemma B.2).

To handle the consensus matrix $C_t^+$, we introduce the **block-maximum norm** on the product space $\mathbb{R}^{N^+d}$:

$$|\mathbf{w}|_{\infty,b} \triangleq \max_{i \in \mathcal{N}^+} \|\mathbf{w}_i\|_2, \quad \text{for } \mathbf{w} = [\mathbf{w}_1^\top, \ldots, \mathbf{w}_{N^+}^\top]^\top.$$

A crucial property of the row-stochastic matrix $C_t^+$ (Proposition B.3) is its **non-expansiveness** with respect to this norm:

$$|(C_t^+ \otimes I_d)\mathbf{w}|_{\infty,b} \leq \max_i \sum_{j \in \mathcal{N}^+} [C_t^+]_{ij}\|\mathbf{w}_j\|_2 \leq \max_i \sum_j [C_t^+]_{ij}|\mathbf{w}|_{\infty,b} = |\mathbf{w}|_{\infty,b}.$$

**Time Window and Scaling.** We proceed by contradiction. Suppose that $\limsup_{t\to\infty} \|\mathbf{w}_t\| = \infty$. Fix a time window $T > 0$. Define a sequence of time indices $\{t_j\}_{j\geq 0}$ with $t_0 = 0$ and

$$t_{j+1} = \min\left\{k > t_j \;\middle|\; \sum_{l=t_j}^{k-1} \alpha_{v,l} \geq T\right\}.$$

For each interval $j$, define the scaling factor $r_j \triangleq \max(1, |\mathbf{w}_{t_j}|_{\infty,b})$. We consider the **scaled iterates** $\hat{\mathbf{w}}_t^j \triangleq \mathbf{w}_t/r_j$ for $t \in [t_j, t_{j+1}]$. Note that $|\hat{\mathbf{w}}_{t_j}^j|_{\infty,b} \leq 1$. Dividing Eq. (27) by $r_j$, the update for the scaled sequence becomes:

$$\hat{\mathbf{w}}_{t+1}^j = (C_t^+ \otimes I_d)\left(\hat{\mathbf{w}}_t^j + \alpha_{v,t}\left[\frac{\mathbf{h}_{\theta_t}(\mathbf{Y}_{t+1})}{r_j} - \mathbf{G}_{\theta_t}(\mathbf{Y}_{t+1})\hat{\mathbf{w}}_t^j + \mathbf{\Xi}_{t+1}\hat{\mathbf{w}}_t^j\right]\right). \tag{28}$$

As $r_j \to \infty$, the driving term $\mathbf{h}/r_j$ vanishes, and the dynamics are dominated by the linear term involving $\mathbf{G}$.

**Reference Trajectory and Tracking.** For each interval $j$, we construct a deterministic, noise-free **reference trajectory** $\{\mathbf{w}_t^{\mathrm{ref},j}\}_{t\geq t_j}$ initialized at $\mathbf{w}_{t_j}^{\mathrm{ref},j} = \hat{\mathbf{w}}_{t_j}^j$:

$$\mathbf{w}_{t+1}^{\mathrm{ref},j} = (C_t^+ \otimes I_d)\left(\mathbf{w}_t^{\mathrm{ref},j} + \alpha_{v,t}\left[\frac{\mathbf{h}(\theta_t)}{r_j} - \mathbf{G}(\theta_t)\mathbf{w}_t^{\mathrm{ref},j}\right]\right). \tag{29}$$

Since the noise terms (centered Poisson fluctuations and martingale differences) have uniform moment bounds (Lemma B.1 and Lemma B.4) and the stepsizes are square-summable, standard stochastic approximation arguments (e.g., Lemma 9 in Konda & Tsitsiklis (2003a)) imply that the scaled sequence tracks the reference trajectory asymptotically:

$$\lim_{j\to\infty} \max_{t_j \leq t \leq t_{j+1}} \left|\hat{\mathbf{w}}_t^j - \mathbf{w}_t^{\mathrm{ref},j}\right|_{\infty,b} = 0 \quad \text{a.s.} \tag{30}$$

**Contraction and Global Boundedness.** We now show that if the iterates grow large (i.e., $r_j \to \infty$), the reference trajectory must contract. By the uniform positive definiteness of $G(\theta)$, there exists $\delta > 0$ such that $\|I - \alpha G(\theta)\|_{\mathrm{op}} \leq 1 - \delta\alpha$ for sufficiently small $\alpha$. Consequently, for the global block-diagonal matrix, $\|I - \alpha\mathbf{G}(\theta)\|_{\mathrm{op}} \leq 1 - \delta\alpha$. Using the non-expansiveness of $(C_t^+ \otimes I_d)$ in the block-maximum norm:

$$|\mathbf{w}_{t+1}^{\mathrm{ref},j}|_{\infty,b} \leq \left|(I - \alpha_{v,t}\mathbf{G}(\theta_t))\,\mathbf{w}_t^{\mathrm{ref},j} + \alpha_{v,t}\frac{\mathbf{h}(\theta_t)}{r_j}\right|_{\infty,b}$$

$$\leq (1 - \delta\alpha_{v,t})|\mathbf{w}_t^{\mathrm{ref},j}|_{\infty,b} + \alpha_{v,t}\frac{K_h}{r_j},$$

where $K_h \triangleq \sup_\theta |\mathbf{h}(\theta)|_{\infty,b}$. Iterating this inequality over the interval $[t_j, t_{j+1}]$ (where $\sum \alpha_{v,k} \approx T$), we obtain:

$$|\mathbf{w}_{t_{j+1}}^{\mathrm{ref},j}|_{\infty,b} \leq e^{-\delta T}|\mathbf{w}_{t_j}^{\mathrm{ref},j}|_{\infty,b} + \frac{C(T)}{r_j} \leq e^{-\delta T} + \frac{C(T)}{r_j}.$$

We choose $T$ large enough such that $e^{-\delta T} \leq 1/4$. Now, consider a subsequence where $r_j \to \infty$. For sufficiently large $j$:

1. The drift term satisfies $C(T)/r_j \leq 1/4$.

2. The tracking error satisfies $\Delta_j \triangleq \max_t |\hat{\mathbf{w}}_t^j - \mathbf{w}_t^{\mathrm{ref},j}|_{\infty,b} \leq 1/4$ (from Eq. 30).

Thus, the scaled iterate at the end of the interval satisfies:

$$|\hat{\mathbf{w}}_{t_{j+1}}^j|_{\infty,b} \leq |\mathbf{w}_{t_{j+1}}^{\mathrm{ref},j}|_{\infty,b} + \Delta_j \leq (1/4 + 1/4) + 1/4 = 3/4.$$

Scaling back to the original variables, $|\mathbf{w}_{t_{j+1}}|_{\infty,b} = r_j|\hat{\mathbf{w}}_{t_{j+1}}^j|_{\infty,b} \leq \frac{3}{4}r_j$. This implies $r_{j+1} = \max(1, |\mathbf{w}_{t_{j+1}}|_{\infty,b}) \leq \max(1, \frac{3}{4}r_j)$. This geometric contraction contradicts the assumption that $r_j$ diverges. Therefore, $\sup_j r_j < \infty$ almost surely, which implies $\sup_{t\geq 0} \|\mathbf{w}_t\| < \infty$ almost surely.

### C.2.2. A SCALED DISAGREEMENT BOUND

In this subsection, we establish a uniform bound on the second moment of the disagreement vector scaled by the step-size. Since the feature mappings are potentially unbounded, we formulate the bound conditioned on the stability of the iterates. This lemma serves as a critical building block for proving the almost sure boundedness of $\mathbf{w}_t$ in the subsequent section.

**Lemma C.1** (Bounded scaled disagreement on stability event). *Suppose Assumptions 3.1– 3.5, 3.7 hold. Let $\mathbf{w}_t^\perp \triangleq (J_\perp \otimes I_d)\mathbf{w}_t$ be the disagreement component of the critic parameters. Define the stability event $\mathcal{E}_t(M) \triangleq \{\sup_{0\leq\tau\leq t}\|\mathbf{w}_\tau\| \leq M\}$. For any $M > 0$, we have:*

$$\sup_{t\geq 0}\mathbb{E}\left[\|\alpha_{v,t}^{-1}\mathbf{w}_t^\perp\|^2\mathbb{I}_{\mathcal{E}_t(M)}\right] < \infty.$$

*Proof.* Recall the global critic update $\mathbf{w}_{t+1} = (C_t^+ \otimes I_d)(\mathbf{w}_t + \alpha_{v,t}\mathbf{y}_{t+1})$, where the aggregated update direction is

$$\mathbf{y}_{t+1} \triangleq h_{\theta_t}(\mathbf{Y}_{t+1}) - G_{\theta_t}(\mathbf{Y}_{t+1})\mathbf{w}_t + \boldsymbol{\xi}_{t+1}. \tag{31}$$

Let $\mathcal{P}_\perp \triangleq J_\perp \otimes I_d$ be the projection operator onto the disagreement subspace. Applying $\mathcal{P}_\perp$ to the update equation, and utilizing the row-stochasticity of $C_t^+$ (which implies $\mathcal{P}_\perp(C_t^+ \otimes I_d) = \mathcal{P}_\perp(C_t^+ \otimes I_d)\mathcal{P}_\perp$), we obtain the recurrence for the disagreement vector:

$$\mathbf{w}_{t+1}^\perp = \mathcal{P}_\perp(C_t^+ \otimes I_d)\bigl(\mathbf{w}_t^\perp + \alpha_{v,t}\mathbf{y}_{t+1}\bigr).$$

Let $\mathbf{q}_t \triangleq \alpha_{v,t}^{-1}\mathbf{w}_t^\perp$ be the scaled disagreement vector. Dividing the recurrence by $\alpha_{v,t+1}$, we have:

$$\mathbf{q}_{t+1} = \frac{\alpha_{v,t}}{\alpha_{v,t+1}}\mathcal{P}_\perp(C_t^+ \otimes I_d)\bigl(\mathbf{q}_t + \mathbf{y}_{t+1}\bigr). \tag{32}$$

We analyze the conditional expectation of $\|\mathbf{q}_{t+1}\|^2$ given $\mathcal{F}_t$. By Proposition B.3, the matrix $\mathbb{E}[(C_t^+)^\top J_\perp C_t^+ \mid \mathcal{F}_t]$ has a spectral radius bounded by $\rho \in (0,1)$. Using Young's inequality $\|a+b\|^2 \leq (1+\gamma)\|a\|^2 + (1+\frac{1}{\gamma})\|b\|^2$ for any $\gamma > 0$, we derive:

$$
\begin{aligned}
\mathbb{E}\left[\|\mathbf{q}_{t+1}\|^2 \mid \mathcal{F}_t\right] &= \left(\frac{\alpha_{v,t}}{\alpha_{v,t+1}}\right)^2 \mathbb{E}\left[\|\mathcal{P}_\perp(C_t^+ \otimes I_d)(\mathbf{q}_t + \mathbf{y}_{t+1})\|^2 \mid \mathcal{F}_t\right] \\
&\leq \left(\frac{\alpha_{v,t}}{\alpha_{v,t+1}}\right)^2 \rho\, \mathbb{E}\left[\|\mathbf{q}_t + \mathbf{y}_{t+1}\|^2 \mid \mathcal{F}_t\right] \\
&\leq \left(\frac{\alpha_{v,t}}{\alpha_{v,t+1}}\right)^2 \rho\left((1+\gamma)\|\mathbf{q}_t\|^2 + (1+\tfrac{1}{\gamma})\mathbb{E}[\|\mathbf{y}_{t+1}\|^2 \mid \mathcal{F}_t]\right).
\end{aligned}
$$

Since $\lim_{t\to\infty} \alpha_{v,t}/\alpha_{v,t+1} = 1$ and $\rho < 1$, we can choose a sufficiently small $\gamma > 0$ and find $t_0$ such that for all $t \geq t_0$:

$$
\lambda_t \triangleq \left(\frac{\alpha_{v,t}}{\alpha_{v,t+1}}\right)^2 \rho(1+\gamma) \leq \frac{1+\rho}{2} \triangleq \bar{\rho} < 1.
$$

For $t \geq t_0$, the recurrence simplifies to:

$$
\mathbb{E}\left[\|\mathbf{q}_{t+1}\|^2 \mid \mathcal{F}_t\right] \leq \bar{\rho}\|\mathbf{q}_t\|^2 + C_\gamma \mathbb{E}\left[\|\mathbf{y}_{t+1}\|^2 \mid \mathcal{F}_t\right], \tag{33}
$$

where $C_\gamma \triangleq \sup_t (\frac{\alpha_{v,t}}{\alpha_{v,t+1}})^2 \rho(1+\frac{1}{\gamma}) < \infty$.

We now control the noise term $\mathbf{y}_{t+1}$ on the event $\mathcal{E}_t(M)$. On this event, $\|\mathbf{w}_t\| \leq M$. Using the definition of $\mathbf{y}_{t+1}$ and the inequality $\|a+b+c\|^2 \leq 3(\|a\|^2 + \|b\|^2 + \|c\|^2)$:

$$
\begin{aligned}
\|\mathbf{y}_{t+1}\|^2 \mathbb{I}_{\mathcal{E}_t(M)} &\leq 3\left(\|h_{\theta_t}(\mathbf{Y}_{t+1})\|^2 + \|G_{\theta_t}(\mathbf{Y}_{t+1})\|_{\mathrm{op}}^2 \|\mathbf{w}_t\|^2 + \|\boldsymbol{\xi}_{t+1}\|^2\right)\mathbb{I}_{\mathcal{E}_t(M)} \\
&\leq 3\left(\|h_{\theta_t}(\mathbf{Y}_{t+1})\|^2 + M^2\|G_{\theta_t}(\mathbf{Y}_{t+1})\|_{\mathrm{op}}^2 + \|\boldsymbol{\xi}_{t+1}\|^2\right).
\end{aligned}
$$

Taking the conditional expectation and applying Lemma B.1 (bounded moments of feature maps) and Lemma B.4 (bounded noise moments), there exists a constant $K(M) < \infty$ independent of $t$ such that:

$$
\mathbb{E}\left[\|\mathbf{y}_{t+1}\|^2 \mid \mathcal{F}_t\right]\mathbb{I}_{\mathcal{E}_t(M)} \leq K(M). \tag{34}
$$

Define $V_t \triangleq \mathbb{E}[\|\mathbf{q}_t\|^2 \mathbb{I}_{\mathcal{E}_t(M)}]$. Note that $\mathcal{E}_{t+1}(M) \subseteq \mathcal{E}_t(M)$ and $\mathcal{E}_t(M) \in \mathcal{F}_t$. Multiplying (33) by $\mathbb{I}_{\mathcal{E}_t(M)}$ and taking total expectations:

$$
\begin{aligned}
V_{t+1} = \mathbb{E}\left[\|\mathbf{q}_{t+1}\|^2 \mathbb{I}_{\mathcal{E}_{t+1}(M)}\right] &\leq \mathbb{E}\left[\|\mathbf{q}_{t+1}\|^2 \mathbb{I}_{\mathcal{E}_t(M)}\right] \\
&= \mathbb{E}\left[\mathbb{E}[\|\mathbf{q}_{t+1}\|^2 \mid \mathcal{F}_t]\mathbb{I}_{\mathcal{E}_t(M)}\right] \\
&\leq \bar{\rho}\, \mathbb{E}\left[\|\mathbf{q}_t\|^2 \mathbb{I}_{\mathcal{E}_t(M)}\right] + C_\gamma \mathbb{E}\left[\mathbb{E}[\|\mathbf{y}_{t+1}\|^2 \mid \mathcal{F}_t]\mathbb{I}_{\mathcal{E}_t(M)}\right] \\
&\leq \bar{\rho} V_t + C_\gamma K(M).
\end{aligned}
$$

This is a linear contraction with a bounded perturbation. By induction, for all $t \geq t_0$:

$$
V_t \leq \bar{\rho}^{t-t_0} V_{t_0} + \frac{C_\gamma K(M)}{1-\bar{\rho}}.
$$

Since $t_0$ is finite and fixed, $V_{t_0} < \infty$. Thus, $\sup_{t \geq 0} V_t < \infty$. $\qquad\square$

### C.2.3. PROOF OF THEOREM 3.10

We define the tracking error of the network-averaged critic parameters as $\hat{e}_t \triangleq \bar{\mathbf{w}}_t - \mathbf{w}^*(\theta_t)$. Recall that the quasi-static fixed point satisfies $\mathbf{w}^*(\theta) = \bar{G}(\theta)^{-1}\bar{h}(\theta)$. Subtracting the update rule of $\mathbf{w}^*(\theta_{t+1})$ from that of $\bar{\mathbf{w}}_{t+1}$ (derived in Step 2), we obtain the error dynamics:

$$
\hat{e}_{t+1} = \left(I - \alpha_{v,t}\bar{G}(\theta_{t+1})\right)\hat{e}_t + \alpha_{v,t}\Delta_t + \xi_{t+1}, \tag{35}
$$

where $\Delta_t$ represents the persistent bias induced by network disagreement and adversarial perturbations:

$$\Delta_t \triangleq \frac{1}{N^+} \left[ \left( \mathbf{1}^\top (C_t^+ - J) \otimes I_d \right) \mathbf{h}(\theta_t) + \left( \mathbf{1}^\top C_t^+ \otimes I_d \right) \alpha_{v,t}^{-1} \mathbf{w}_t^{+\perp} \right].$$

The term $\xi_{t+1} \triangleq \varepsilon_{t+1}^{(1)} + \varepsilon_{t+1}^{(2)}$ aggregates the martingale difference noise, Markovian sampling noise, and the drift due to the slowly evolving policy $\theta_t$:

$$\varepsilon_{t+1}^{(1)} = \alpha_{v,t} \bar{G}(\theta_{t+1}) \left[ \bar{h}_{\theta_t}(\mathbf{Y}_{t+1}) - \bar{h}(\theta_t) - \left( \bar{G}_{\theta_t}(\mathbf{Y}_{t+1}) - \bar{G}(\theta_t) \right) \bar{\mathbf{w}}_t \right]$$
$$+ \frac{\alpha_{v,t}}{N^+} \bar{G}(\theta_{t+1})(\mathbf{1}^\top (C_t^+ - J) \otimes I_d) \left( h_{\theta_t}(\mathbf{Y}_{t+1}) - h(\theta_t) \right) + \alpha_{v,t} \bar{G}(\theta_{t+1}) \zeta_{t+1},$$
$$\varepsilon_{t+1}^{(2)} = \left( \bar{G}(\theta_{t+1}) - \bar{G}(\theta_t) \right) \bar{\mathbf{w}}_t - \left( \mathbf{w}^*(\theta_{t+1}) - \mathbf{w}^*(\theta_t) \right).$$

**Analysis of Noise Terms:** The term $\varepsilon_{t+1}^{(1)}$ consists of martingale differences sequences with bounded moments scaled by $\alpha_{v,t}$. The term $\varepsilon_{t+1}^{(2)}$ represents the tracking drift on the slower timescale. Since the maps $\theta \mapsto \mathbf{w}^*(\theta)$ and $\theta \mapsto \bar{G}(\theta)$ are Lipschitz (Lemma B.1) and $\|\theta_{t+1} - \theta_t\| = O(\alpha_{\theta,t})$, we have $\|\varepsilon_{t+1}^{(2)}\| = O(\alpha_{\theta,t})$. By the timescale separation assumption $\alpha_{\theta,t} = o(\alpha_{v,t})$, it follows that $\|\xi_{t+1}\| = o(\alpha_{v,t})$ almost surely.

**Comparison Recursion:** To analyze the asymptotic behavior of $\hat{e}_t$, we employ the **comparison argument** (Borkar, 2008). Let $T > 0$ be a fixed time window. Define a sequence of time indices $\{t_j\}_{j \geq 0}$ such that $t_0 = 0$ and $t_{j+1} = \min\{t > t_j \mid \sum_{k=t_j}^{t-1} \alpha_{v,k} \geq T\}$. For each interval $j$, we construct a deterministic comparison sequence $\{e_t^j\}_{t \geq t_j}$ initialized at $e_{t_j}^j = \hat{e}_{t_j}$ and evolving as:

$$e_{t+1}^j = (I - \alpha_{v,t} \bar{G}(\theta_{t+1})) e_t^j + \alpha_{v,t} \Delta_t. \tag{36}$$

Invoking the standard tracking lemma for stochastic approximation (e.g., Lemma 1 in Chapter 2 of Borkar (2008) or Lemma 15 in Konda & Tsitsiklis (2003a)), the trajectory of $\hat{e}_t$ asymptotically tracks $e_t^j$ as the cumulative noise vanishes:

$$\lim_{j \to \infty} \max_{t_j \leq t \leq t_{j+1}} \|\hat{e}_t - e_t^j\| = 0 \quad \text{a.s.} \tag{37}$$

Thus, it suffices to bound the limit superior of $\|e_t^j\|$. Taking the norm of the recursion and using the uniform positive definiteness of $\bar{G}(\theta)$, for sufficiently large $t$, we have $\|I - \alpha_{v,t} \bar{G}(\theta_{t+1})\| \leq 1 - \frac{\delta}{2} \alpha_{v,t}$. Thus:

$$\|e_{t+1}^j\| \leq \left( 1 - \frac{\delta}{2} \alpha_{v,t} \right) \|e_t^j\| + \alpha_{v,t} \|\Delta_t\|.$$

Define the discrete state transition product $\Phi(m,t) \triangleq \prod_{k=m}^{t-1} (1 - \frac{\delta}{2} \alpha_{v,k})$ for $t > m$, with $\Phi(t,t) = 1$. Unrolling the recurrence from any $m \in [t_j, t)$:

$$\|e_t^j\| \leq \Phi(m,t) \|e_m^j\| + \sum_{k=m}^{t-1} \alpha_{v,k} \|\Delta_k\| \Phi(k+1,t). \tag{38}$$

Let $\bar{\Delta}_m \triangleq \sup_{k \geq m} \|\Delta_k\|$ denote the tail supremum of the perturbation. We can bound the summation term using a telescoping sum argument. Observe that $\alpha_{v,k} \Phi(k+1,t) = \frac{2}{\delta} \left[ \Phi(k+1,t) - (1 - \frac{\delta}{2} \alpha_{v,k}) \Phi(k+1,t) \right] = \frac{2}{\delta} [\Phi(k+1,t) - \Phi(k,t)]$. Thus:

$$\sum_{k=m}^{t-1} \alpha_{v,k} \|\Delta_k\| \Phi(k+1,t) \leq \bar{\Delta}_m \sum_{k=m}^{t-1} \frac{2}{\delta} \left( \Phi(k+1,t) - \Phi(k,t) \right)$$
$$= \frac{2\bar{\Delta}_m}{\delta} \left( \Phi(t,t) - \Phi(m,t) \right) \leq \frac{2\bar{\Delta}_m}{\delta}.$$

Substituting this back into (38), we have $\|e_t^j\| \leq \Phi(m,t) \|e_m^j\| + \frac{2\bar{\Delta}_m}{\delta}$. Since $\sum \alpha_{v,t} = \infty$, $\Phi(m,t) \to 0$ as $t \to \infty$. Therefore, for any fixed $m$, $\limsup_{t \to \infty} \|e_t^j\| \leq \frac{2\bar{\Delta}_m}{\delta}$. Combining this with (37), we obtain $\limsup_{t \to \infty} \|\hat{e}_t\| \leq \frac{2\bar{\Delta}_m}{\delta}$ almost surely. Finally, letting $m \to \infty$, we have $\bar{\Delta}_m \downarrow \limsup_{t \to \infty} \|\Delta_t\|$. We conclude:

$$\limsup_{t \to \infty} \|\bar{\mathbf{w}}_t - \mathbf{w}^*(\theta_t)\| \leq \frac{2}{\delta} \limsup_{t \to \infty} \|\Delta_t\| \quad \text{a.s.} \tag{39}$$

Since $\|\mathbf{w}_{i,t} - \bar{\mathbf{w}}_t\| \to 0$ by Step 1 (Corollary B.1), the same bound applies to each cooperative agent, establishing (15).

**Special Case ($H = 0$).** In the absence of Byzantine agents, the consensus matrix $C_t^+$ corresponds to the full cooperative graph and is doubly stochastic in expectation. This implies $\mathbf{1}^\top (C_t^+ - J) = 0$. Furthermore, the disagreement term involving $\mathbf{w}_t^{+\perp}$ vanishes asymptotically. Consequently, $\lim_{t\to\infty} \Delta_t = 0$ a.s. The error recursion reduces to a standard contraction driven by vanishing noise, yielding:

$$\lim_{t\to\infty} \|\mathbf{w}_{i,t} - \mathbf{w}^*(\theta_t)\| = 0 \quad \text{a.s.}$$

This recovers the standard convergence result for decentralized actor-critic algorithms without adversaries.

### C.3. Proof of Theorem 3.11

We analyze the convergence of the actor updates using the general framework established in Appendix B.2.

We first recall the definition $\mathcal{F}_t = \sigma(\boldsymbol{\theta}_0, \mathbf{w}_0, s_0, \ldots, \boldsymbol{\theta}_t, \mathbf{w}_t, s_t)$ be the filtration representing the entire history. The actor update for agent $i \in \mathcal{N}^+$ is given by:

$$\theta_{t+1}^i = \Psi_{\Theta^i} \left( \theta_t^i + \alpha_{\theta,t} \cdot \hat{Y}_{t+1}^i \right), \tag{40}$$

where $\hat{Y}_{t+1}^i$ is the stochastic update direction computed from the transition sample $(s_t, a_t, s_{t+1})$. To map this to the general form (22), we explicitly define the sample-based update function $\hat{F}^i$:

$$\hat{F}^i(s, a, s'; \boldsymbol{\theta}, \mathbf{w}) \triangleq \left( \psi(s,a)^\top \omega + \gamma \phi(s')^\top v - \phi(s)^\top v \right) \cdot \nabla_{\theta^i} \log \pi^i(a^i \mid s; \theta^i). \tag{41}$$

And we define $F^i(s, a; \cdot) = \mathbb{E}_{s' \sim P(\cdot|s,a)}[\hat{F}^i(s, a, s'; \cdot)]$.

We define the instantaneous mean field $g_t^i(\boldsymbol{\theta})$ as the expected update direction conditioned on the current critic parameters $\mathbf{w}_t^i$, evaluated over the stationary distribution $\eta_{\boldsymbol{\theta}}$:

$$g_t^i(\boldsymbol{\theta}) \triangleq g^i(\boldsymbol{\theta}, \mathbf{w}_t^i) = \mathbb{E}_{(s,a)\sim\eta_{\boldsymbol{\theta}}} \left[ F^i(s, a; \boldsymbol{\theta}, \mathbf{w}_t^i) \right]. \tag{42}$$

The stochastic update $\hat{Y}_{t+1}^i = \hat{F}^i(s_t, a_t, s_{t+1}; \boldsymbol{\theta}_t, \mathbf{w}_t^i)$ is then decomposed into three terms:

$$\hat{Y}_{t+1}^i = g_t^i(\boldsymbol{\theta}_t) + \delta M_{t+1}^i + \beta_t^i, \tag{43}$$

where the noise terms are identified as:

- **Martingale Difference $\delta M_{t+1}^i$:** This term captures the noise from sampling the action $a_t$ and the next state $s_{t+1}$:

$$\delta M_{t+1}^i \triangleq \hat{Y}_{t+1}^i - \mathbb{E}[\hat{Y}_{t+1}^i \mid \mathcal{F}_t].$$

- **Markovian Bias $\beta_t^i$:** This term arises from the discrepancy between the conditional expectation (started at $s_t$) and the stationary expectation:

$$\beta_t^i \triangleq \mathbb{E}[\hat{Y}_{t+1}^i \mid \mathcal{F}_t] - g_t^i(\boldsymbol{\theta}_t).$$

We now verify the conditions required by Theorem B.5:

**1) Boundedness:** The actor parameters $\theta_t^i$ are bounded by the projection $\Psi_{\Theta^i}$. By Theorem 3.10, the critic parameters $\mathbf{w}_t^i$ are bounded almost surely. Since the features $\phi, \psi$ and score functions belong to class $\mathcal{D}$ (Assumptions 3.3-3.4), they possess bounded moments. Thus, the update terms are bounded in $L_2$ and almost surely.

**2) Stepsizes:** Assumption 3.5 ensures $\alpha_{\theta,t}$ satisfies the Robbins-Monro conditions.

**3) Martingale Noise:** By construction, $\mathbb{E}[\delta M_{t+1}^i \mid \mathcal{F}_t] = 0$. Due to the boundedness of parameters and the polynomial growth of features, $\sup_t \mathbb{E}[\|\delta M_{t+1}^i\|^2 \mid \mathcal{F}_t] < \infty$ a.s.

**4) Vanishing Bias:** Under Assumption 3.3 (Uniform Geometric Ergodicity) and the two-timescale separation ($\|\theta_{t+1} - \theta_t\| \to 0$), standard Poisson equation arguments for stochastic approximation (Konda & Tsitsiklis, 2003b, Lemma 6.1) guarantee that the cumulative bias is negligible, implying $\beta_t^i \to 0$ almost surely.

**5) Regularity and Limit Set:** We verify the uniform continuity of $g_t^i(\boldsymbol{\theta})$ and the USC property of the limit set via the following lemmas.

**Lemma C.2** (Continuity of Stationary Expectations). *Suppose Assumption 3.3 holds. Let $\{h(\cdot; \boldsymbol{\theta})\}_{\boldsymbol{\theta} \in \boldsymbol{\Theta}}$ be a family of functions satisfying: (i) Uniform growth in $\mathcal{D}$; (ii) Finite-step continuity (the map $\boldsymbol{\theta} \mapsto \mathbb{E}_{\boldsymbol{\theta}, s}[h(S_k, A_k; \boldsymbol{\theta})]$ is continuous for fixed $k$). Then, the map $\boldsymbol{\theta} \mapsto \mathbb{E}_{\eta_{\boldsymbol{\theta}}}[h(\cdot; \boldsymbol{\theta})]$ is continuous.*

*Proof.* Let $\boldsymbol{\theta}, \boldsymbol{\theta}' \in \boldsymbol{\Theta}$. Decompose the error using a finite-horizon approximation $k$:

$$|\mathbb{E}_{\eta_{\boldsymbol{\theta}}} - \mathbb{E}_{\eta_{\boldsymbol{\theta}'}}| \leq \underbrace{|\mathbb{E}_{\eta_{\boldsymbol{\theta}}} - \mathbb{E}_{k, \boldsymbol{\theta}}|}_{\text{(I)}} + \underbrace{|\mathbb{E}_{k, \boldsymbol{\theta}} - \mathbb{E}_{k, \boldsymbol{\theta}'}|}_{\text{(II)}} + \underbrace{|\mathbb{E}_{k, \boldsymbol{\theta}'} - \mathbb{E}_{\eta_{\boldsymbol{\theta}'}}|}_{\text{(III)}}.$$

Terms (I) and (III) are bounded by $C\rho^k L(s)$ due to uniform geometric ergodicity (Lemma 4.3(e) in Konda & Tsitsiklis (2003b)). Choosing $k$ large enough makes these $< \epsilon$. For fixed $k$, term (II) vanishes as $\|\boldsymbol{\theta} - \boldsymbol{\theta}'\| \to 0$ by the Dominated Convergence Theorem, since transitions and policies are continuous. □

**Lemma C.3** (Joint Continuity of Mean Field). *Under Assumptions 1 and 2, the deterministic function $\bar{f}^i(\boldsymbol{\theta}, \mathbf{w}) \triangleq \mathbb{E}_{\eta_{\boldsymbol{\theta}}}[F^i(\cdot; \boldsymbol{\theta}, \mathbf{w})]$ is jointly continuous in $(\boldsymbol{\theta}, \mathbf{w})$.*

*Proof.* The integrand $F^i$ belongs to class $\mathcal{D}$ uniformly for bounded $\mathbf{w}$. For fixed $\mathbf{w}$, $\boldsymbol{\theta} \mapsto \bar{f}^i(\boldsymbol{\theta}, \mathbf{w})$ is continuous by Lemma C.2. Since $F^i$ is linear in $\mathbf{w}$ with coefficients (stationary feature expectations) bounded on compact sets, $\bar{f}^i$ is Lipschitz continuous with respect to $\mathbf{w}$. Thus, it is jointly continuous. □

**Proposition C.4** (Uniform Continuity). *The sequence of functions $\{g_t^i(\cdot)\}_{t \geq 0}$ is equicontinuous. Specifically, $g_t^i(\boldsymbol{\theta})$ is continuous in $\boldsymbol{\theta}$ uniformly in $t$.*

*Proof.* By Theorem 3.10, there exists a compact set $\mathcal{W}$ such that $\mathbf{w}_t^i \in \mathcal{W}$ for all $t$ almost surely. Since $\bar{f}^i$ is jointly continuous on the compact product space $\boldsymbol{\Theta} \times \mathcal{W}$ (Lemma C.3), it is uniformly continuous. Thus, for any $\epsilon > 0$, there exists $\delta > 0$ independent of $t$ and $\mathbf{w} \in \mathcal{W}$ such that $\|\boldsymbol{\theta} - \boldsymbol{\theta}'\| < \delta \implies \|g^i(\boldsymbol{\theta}, \mathbf{w}) - g^i(\boldsymbol{\theta}', \mathbf{w})\| < \epsilon$. □

Finally, we define the limit set-valued map $\bar{g}(\boldsymbol{\theta})$ as in Theorem B.5. Since $\{g_t^i\}$ is uniformly continuous, $\bar{g}(\boldsymbol{\theta})$ is Upper Semicontinuous.

All conditions of Theorem B.5 are verified. Thus, the actor parameters $\theta_t^i$ converge almost surely to the limit set of the differential inclusion $\dot{\theta}^i \in \Psi_{\Theta^i}[\bar{g}^i(\boldsymbol{\theta})]$. □

# D. Detailed Case Studies of Service Systems with Affinity

### D.1. Details of case 1: Large language model semantic routing

D.1.1. MODEL

We instantiate the general service system with affinity introduced in Section 2 to model semantic routing for large language model (LLM) inference.

*System structure.* A set of routers $\mathcal{N}_t$ generate and forward inference requests, and a set of parallel GPU servers $\mathcal{N}_s$ execute LLM inference. The directed graph $\mathcal{G}$ is bipartite, where each router is connected to all GPU servers. Each request is associated with a semantic task type $\tau \in \{1, \ldots, K\}$, representing coarse semantic categories such as intent or domain.

*Traffic state.* The traffic state $x$ corresponds to queue backlogs at all routers and GPU servers. Time is slotted. At each time step, each router independently generates a new request according to a Bernoulli process with arrival probability $p$. Routers and GPU servers maintain FIFO queues of individual requests.

*Affinity state.* Each GPU server $j \in \mathcal{N}_s$ maintains a semantic memory vector

$$\mathbf{h}_j(t) \in \mathbb{R}_{\geq 0}^K,$$

where $h_{j,k}(t)$ counts how many requests of task type $k$ have recently been processed by server $j$. This memory models the content of the GPU's KV cache.

The affinity between a router request of task type $\tau$ and GPU server $j$ is defined as

$$f_{ij}(t) \triangleq \frac{h_{j,\tau}(t)}{\sum_{k=1}^{K} h_{j,k}(t) + \varepsilon},$$

which takes values in $[0, 1]$. A larger affinity indicates that the server's KV cache is better aligned with the incoming request, leading to faster effective service.

*Service dynamics.* For a GPU server $j$, the effective service rate for a request of task type $\tau$ is affinity-aware:

$$\mu_j^{\mathrm{eff}}(\tau, t) = \mu_j\big(1 + \beta f_{ij}(t)\big),$$

where $\mu_j$ is the nominal service rate and $\beta \geq 0$ controls the strength of semantic acceleration. Upon successful service completion, the corresponding semantic memory entry $h_{j,\tau}$ is incremented and subject to mild exponential decay to avoid unbounded growth.

*State space.* The overall system state is

$$s(t) = \big(x(t), f(t)\big),$$

which matches the general formulation in Section 2 with a continuous affinity state and an unbounded queueing state.

*Agents and actions.* Routers are the learning agents of interest. At each time step, a router selects one downstream GPU server to forward its processed requests. GPU servers execute a fixed affinity-aware service policy and update their semantic memories locally.

*Observations.* Each router $i$ observes a local observation vector $o_i(t)$ including its own queue length, the queue lengths of all GPU servers, and the affinity values between its head-of-line request and each GPU server. This corresponds to a decentralized partially observed MMDP consistent with the general model.

*Reward design.* The instantaneous reward of router $i$ is defined as a function of the global state and its local action,

$$r_i\big(s, a_i\big) = -w_1 x_i - w_2\big(x_{a_i} - \bar{x}_{\mathrm{s}}\big)^2 + w_3 f_{ia_i},$$

where $\bar{x}_{\mathrm{s}}$ denotes the average queue length across GPU servers, and $w_1, w_2, w_3 > 0$ are weighting coefficients. The reward penalizes router congestion, enforces load balancing, and promotes affinity-aware routing.

### D.1.2. ALGORITHM

We apply the secure decentralized actor–critic framework described in Section 2.3 *Basis function.* Given a local observation $o_i(t)$, we construct a feature vector $\phi_i(t) \in \mathbb{R}^d$ that encodes queueing and affinity information.

Let $s_i(t) \in \mathbb{R}^{d_0}$ denote the raw state variables extracted from the observation, including the router queue length, downstream GPU queue lengths, and affinity values. To prevent numerical instability caused by unbounded semantic memory states, we apply a logarithmic transformation

$$\tilde{s}_i(t) = \log\big(1 + |s_i(t)|\big),$$

which ensures bounded growth and preserves relative ordering.

The final feature vector consists of three components:

$$\phi_i(t) = \big[1, \ \tilde{s}_i(t), \ \mathrm{vec}\big(\tilde{s}_i(t)\tilde{s}_i(t)^{\top}\big)\big],$$

where only the upper-triangular elements of the quadratic term are retained. The quadratic features capture second-order interactions between queue congestion and semantic affinity, enabling the policy to represent decisions such as preferring a lightly loaded GPU server when semantic alignment is high.

*Policy mixture and safety mechanism.* To ensure stability under adversarial conditions and severe congestion, each router deploys a mixture of a learned policy and a predefined safe policy.

The safe policy $\pi^{\mathrm{safe}}$ is chosen as a uniform routing policy over downstream GPU servers, corresponding to randomized load balancing. This policy is stateless, decentralizable, and guarantees bounded queue growth under mild load conditions.

At each time step, router $i$ forms a mixed policy

$$\pi_i^{\mathrm{mix}}(\cdot \mid o_i) = \alpha_i(t)\,\pi_{\theta_i}(\cdot \mid o_i) + \big(1 - \alpha_i(t)\big)\,\pi^{\mathrm{safe}}(\cdot),$$

where the mixing coefficient $\alpha_i(t) \in [0,1]$ is determined adaptively using the critic value estimate

$$\alpha_i(t) = \min\left(1,\ \frac{C_0}{|V_{w_i}(o_i(t))| + \varepsilon}\right).$$

A large value estimate indicates potential congestion or instability, in which case the agent relies more heavily on the safe policy. When the estimated value is small, the learned policy dominates.

To maintain decentralized consistency, routers optionally share a common random coin to synchronize the mixture decision, ensuring coordinated switching between learned and safe behaviors. Actions are sampled from the deployed mixed policy $\pi_i^{\mathrm{mix}}$. The critic is updated via temporal-difference learning. The actor is updated using a corrected policy gradient that accounts for sampling from the mixture distribution. Specifically, the policy gradient is weighted by an importance factor proportional to the contribution of the learned policy in the mixture, ensuring unbiased optimization of $\pi_{\theta_i}$ while preserving the safety guarantees of deployment.

### D.1.3. EXPERIMENT

*Setup.* We consider a system with 5 routers and 3 GPU servers. The number of semantic task types is $K = 3$. Request arrivals follow independent Bernoulli processes with probability $p=0.8$, corresponding to an average arrival rate of 0.8 tasks/s per router. The nominal service rates are set to $\mu_{\mathrm{t}} = 1$ task/s for routers and $\mu_{\mathrm{s}} = 1$ task/s for GPU servers. *Learned affinity structure.* Figure 7 visualizes the learned task–server affinity using heatmap representations under three conditions: no attack, Byzantine attack without defense, and attack with the proposed resilient mechanism enabled.

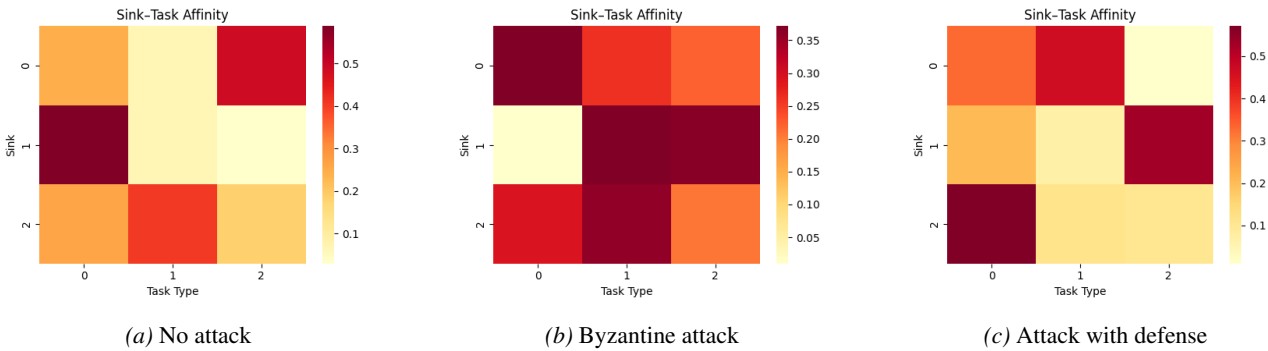

*(a)* No attack   *(b)* Byzantine attack   *(c)* Attack with defense

*Figure 7.* Learned task–server affinity heatmaps under different conditions. Without attacks or with the proposed defense enabled, distinct server preferences emerge for specific task types. In contrast, adversarial manipulation without defense leads to diffuse and unstructured affinities.

In the no-attack setting, the affinity heatmap exhibits clear structure: specific GPU servers develop strong preferences for particular semantic task types, as indicated by concentrated high-intensity regions. This reflects the emergence of implicit server specialization driven by KV-cache reuse and affinity-aware routing.

Under Byzantine attack without defense, the affinity patterns become significantly more diffuse. Adversarial perturbations distort routing decisions and contaminate the semantic memory at GPU servers, preventing the formation of stable task–server associations. As a result, the learned affinity fails to capture meaningful specialization, leading to inefficient routing and increased congestion.

When the proposed defense mechanism is enabled, the structured affinity patterns are largely recovered despite the presence of adversarial nodes. The policy mixture mechanism suppresses unsafe routing behaviors during high-congestion or manipulated states, allowing the system to revert to stable load balancing and gradually re-establish coherent task–sink affinities. This demonstrates that the proposed approach not only stabilizes queue dynamics but also preserves semantic specialization under adversarial conditions.

## D.2. Details of case 2: Distributed polling for edge computing

### D.2.1. MODEL

We consider a distributed polling system motivated by edge computing architectures, where a set of geographically distributed data sources generate computation tasks, and a set of edge servers dynamically poll sources to provide service. *System Structure* The system is modeled as a directed bipartite graph $\mathcal{G} = \langle \mathcal{N}, \mathcal{E} \rangle$, where the node set $\mathcal{N} = \mathcal{N}_s \cup \mathcal{N}_e$ consists of *sources* $\mathcal{N}_s$ and *edge servers* $\mathcal{N}_e$. Each directed edge $(i, j) \in \mathcal{E}$ represents that server $j$ can poll source $i$ for tasks. We assume a fully connected polling topology, i.e., every server can poll every source.

*Traffic state.* Each source $i \in \mathcal{N}_s$ maintains a queue with length $x_i(t) \in \mathbb{Z}_{\geq 0}$. Jobs arrive at router $i$ according to an independent Bernoulli process with arrival probability $p$. Edge servers do not maintain persistent queues; instead, polling and service are combined into a single operation.

The global traffic state is $x(t) = [x_1(t), \ldots, x_{|\mathcal{N}_s|}(t)]^\top$.

*Affinity state.* For each server $j \in \mathcal{N}_e$, we define an implicit, time-varying *affinity state* $f_j(t)$ capturing the server's current polling preference over sources. In practice, this affinity is reflected by the server's selected scheduling policy (e.g., longest-queue-first, shortest-queue-first, round-robin, or random polling), as well as its most recently polled source. Switching the effective affinity induces a non-negligible switch-over delay. *System dynamics.* Given the joint action $a(t)$, each server polls a source according to its selected policy. If a server switches its polled source relative to the previous step, a switch-over delay is incurred, during which no service is provided. Otherwise, the server serves up to $\mu_j$ jobs from the chosen source. The resulting dynamics are Markovian and define a transition kernel $P((s, a), \cdot)$.

*State space.* The overall system state is $s(t) = \big(x(t), f(t)\big)$, where $f(t)$ encodes the servers' polling preferences and switch-over timers. The state space is therefore $\mathcal{S} = \mathbb{Z}_{\geq 0}^{|\mathcal{N}_s|} \times \mathcal{F}$, where $\mathcal{F}$ is a finite-dimensional space representing policy selections and switch states.

*Agents and actions.* Each edge server $j \in \mathcal{N}_e$ acts as an agent. At each time step, server $j$ selects an action $a_j(t) \in \mathcal{A}_j$, where $\mathcal{A}_j$ corresponds to a discrete set of polling policies, e.g., longest-queue-first (LQF), shortest-queue-first (SQF), round-robin (RR), and random polling (RND). Sources do not take actions.

*Reward design.* The instantaneous reward of server $i$ is defined as a function of the state and its local action,

$$r_i\big(s, a_i\big) = -w_1 tanh\left(\frac{\sum_k x_k}{Q_0}\right) - w_2 \mathbf{1}[a_i == switch]\,,$$

where $w_1, w_2 > 0$ are weighting coefficients. The first term penalizes global congestion via queue pressure, and the last term discourages wasted polling actions. The cooperative objective is to minimize the team-average long-term cost, as defined in Section 2.3.

### D.2.2. ALGORITHM

We adopt the same secure decentralized actor–critic framework as in Case 1, with linear function approximation.

*Basis function.* For each server, the local observation is mapped to a feature vector $\phi_i(t)$ using a fixed nonlinear basis. Let $s_i(t)$ denote the raw state variables extracted from the observation, including queue lengths of all sources. To prevent numerical instability, we apply a logarithmic transformation

$$\tilde{s}_i(t) = \log\big(1 + |s_i(t)|\big).$$

The feature vector is constructed as

$$\phi_i(t) = \big[1, \ \tilde{s}_i(t), \ \text{vec}\big(\tilde{s}_i(t)\tilde{s}_i(t)^\top\big)\big],$$

where only upper-triangular quadratic terms are retained. This basis enables the policy to capture interactions between different queue backlogs, which is critical for coordinated polling decisions.

As in Case 1, a safety-aware policy mixture is employed at execution time, combining the learned policy with a predefined safe polling policy to ensure stability under adversarial conditions. We omit details here for brevity.

### D.2.3. EXPERIMENT

*Setup.* We consider a polling system with 6 routers and 10 servers. Time is slotted with one-second intervals. Each source generates jobs according to an independent Bernoulli process with probability $p = 0.8$. Each server can process at most one job per second.

*Delay time performance.*

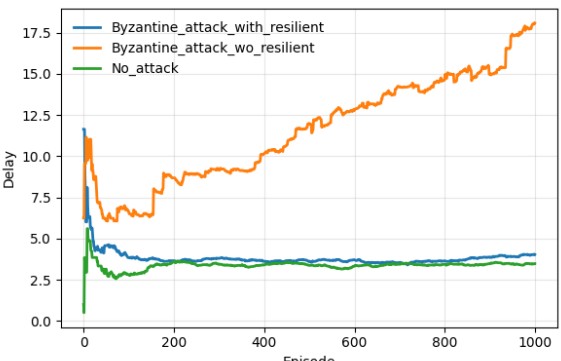

*Figure 8.* Average job waiting time across all queues during training. Waiting times remain bounded under no attack and attack-with-defense settings, but increase significantly under Byzantine attacks without protection.

Figure 8 illustrates the average job waiting time across all queues. Consistent with queue-length dynamics, waiting time remains bounded and converges under both the no-attack and attack-with-defense settings. In contrast, Byzantine attacks without protection lead to rapidly increasing delays.

These results confirm that stabilizing polling decisions via the proposed resilient learning mechanism directly translates into improved service quality. In particular, mitigating adversarial influence prevents excessive queue pressure, thereby reducing delay under realistic polling and switching overheads.

### D.3. Details of case 3: Delivery for smart manufacturing

### D.3.1. MODEL

We consider a smart manufacturing delivery system, where multiple mobile agents transport tasks from warehouses to destinations on a factory floor, subject to stochastic task arrivals, limited battery capacity, and charging constraints.

*System structure* The manufacturing floor is modeled as a 2D grid graph $\mathcal{G} = \langle \mathcal{V}, \mathcal{E} \rangle$, where each node represents a physical location and edges correspond to feasible movements between adjacent cells. Shortest-path distances are defined with respect to this grid topology.

*Demands.* Each demand $i \in \mathcal{D}$ is defined by a warehouse location $w_i \in \mathcal{V}$ and a destination $d_i \in \mathcal{V}$. Tasks arrive to demand $i$ according to an independent Bernoulli process with probability $p$. Arrived tasks are stored in a FIFO queue, recording their arrival times.

*Agents.* A set of mobile delivery agents $\mathcal{A}$ operates on the grid. Each agent $i \in \mathcal{A}$ has: (i) a current position $p_i(t) \in \mathcal{V}$, (ii) a discrete operational stage $\sigma_i(t)$ (idle, traveling to warehouse, delivering, charging), and (iii) a battery level $b_i(t) \in [0, b_{\max}]$.

Agents consume energy when moving and must visit designated charging stations once the battery level drops below a threshold.

*State space.* The global system state is defined as

$$s(t) = \big(q(t),\ p(t),\ \sigma(t),\ b(t)\big),$$

where $q(t)$ denotes the vector of demand queue lengths, $p(t)$ agent positions, $\sigma(t)$ agent stages, and $b(t)$ normalized battery levels. The state space is finite but large due to the combinatorial coupling between queues and agent locations.

*Actions.* Each agent acts independently. When idle, agent $i$ selects a demand index $a_i(t) \in \{1, \ldots, |\mathcal{D}|\}$, corresponding to committing to serve that demand. Movement, delivery, and charging behaviors then follow deterministic rules based on shortest paths and battery constraints.

*System dynamics.* Once an agent commits to a demand, it first travels to the corresponding warehouse, then transports tasks to the destination, serving up to $\mu$ tasks per visit. If the battery level falls below a predefined threshold, the agent is forced to reroute to the nearest charging station. Unfinished tasks remain queued and accumulate waiting time.

The resulting dynamics form a controlled Markov process with stochastic arrivals and deterministic motion primitives.

*Reward design.* The instantaneous reward of agent $i$ is designed to balance throughput and system stability:

$$r_i(s, a_i) = w_1 \, R_i^{\text{comp}}(s, a_i) - w_2 \, R_i^{\text{queue}}(s, a_i) - w_3 \, R_i^{\text{move}}(s, a_i),$$

where each term captures a distinct operational objective.

The task completion reward

$$R_i^{\text{comp}}(s, a_i)$$

denotes the number of tasks successfully delivered by agent $i$ at time $t$ as a direct consequence of action $a_i$. This term promotes throughput and timely task fulfillment.

To penalize congestion while filtering stochastic fluctuations, we define the queue-related penalty using a smoothed deviation of total queue length. Let

$$Q = \sum_{i \in \mathcal{D}} q_i$$

denote the total number of outstanding tasks in the system, which is a deterministic function of the state $s$. We further define an exponentially smoothed reference level

$$\bar{Q} = (1 - \eta) \, \bar{Q}(t - 1) + \eta \, Q, \qquad \eta \in (0, 1),$$

which captures the long-term average congestion. The queue penalty term is then defined as

$$R_i^{\text{queue}}(s, a_i) = Q - \bar{Q},$$

penalizing actions taken under increasing congestion while avoiding overreaction to short-term noise.

Finally, the movement cost

$$R_i^{\text{move}}(s, a_i)$$

captures energy consumption induced by action $a_i$. Specifically, it is proportional to the distance traveled by agent $i$ under the chosen action and the corresponding battery expenditure, both of which are deterministic functions of the current position, battery level, and action selection.

This reward structure encourages agents to complete tasks efficiently while stabilizing system-wide queues and respecting energy constraints, thereby minimizing long-term waiting time and preventing queue explosion.

### D.3.2. ALGORITHM

We adopt a similar secure decentralized actor–critic framework as in Case 1 and Case 2, with linear function approximation. *Basis function.* Each agent maps its local observation to a feature vector $\phi_j(t)$. Let $s_j(t)$ denote the raw state variables extracted from the observation, including demand queue lengths, agent positions, battery levels, etc. The feature vector is constructed as

$$\phi_j(t) = \left[ 1, \; s_j(t), \; \text{vec}\big(s_j(t) s_j(t)^\top\big)_{\text{upper-triangular}} \right].$$

Unlike the previous cases, no logarithmic transformation is applied; all linear and quadratic interactions of the raw state are retained. This allows the policy to capture both first-order and pairwise relationships between queues, positions, and battery levels.

*Policy mixture.* To ensure safety under high congestion or adversarial conditions, we mix the learned policy $\pi_{\theta_j}$ with a safe policy $\pi_j^{\text{safe}}$ that deterministically selects the demand queue with the largest backlog. The mixing factor $\alpha \in [0, 1]$

is determined from the absolute value of the current value estimate $V_{w_j}(o_j)$: high value magnitude (indicating unsafe / congested state) leads to smaller $\alpha$, favoring the safe policy. The deployed policy is

$$\pi_j^{\text{mix}} = \alpha\,\pi_{\theta_j} + (1 - \alpha)\,\pi_j^{\text{safe}},$$

from which the action is sampled at each time step. The critic is updated using standard TD error, and the actor is updated via policy gradient with importance sampling weights to account for the mixture sampling distribution.

### D.3.3. EXPERIMENT

*Setup.* We simulate a smart manufacturing floor with multiple warehouse–destination pairs and mobile delivery agents operating on a grid. Task arrivals follow independent Bernoulli processes with probability $p = 0.8$. Agents have finite battery capacities and must periodically recharge. All results are averaged over multiple random seeds.

