# OpenReview forum: "Secure Multi-agent Reinforcement Learning for Service Systems with Affinity and Byzantine Nodes: Stability Analysis and Protection Design"
_ICML.cc/2026/Conference — ICML 2026 regular_

### Official Review · Reviewer_Lw4w · 2026-02-17

**Soundness:** 2
**Presentation:** 2
**Significance:** 2
**Originality:** 2
**Overall Recommendation:** 2
**Confidence:** 3

**Summary:**

The paper tackles decentralized multi-agent reinforcement learning for networked service systems with time-varying job-server affinity and unbounded queue states under Byzantine (malicious) agents. It proposes a resilient decentralized actor-critic method that combines coordinate-wise W-MSR trimmed aggregation on exchanged parameters with a synchronized ''safe-policy'' mixture (mixing learned policies with a stabilizing baseline) to keep learning stable.
It provides stability/convergence guarantees (convergence to a bounded neighborhood whose size reflects adversarial impact) and demonstrates the approach on LLM semantic routing, edge computing polling, and smart manufacturing delivery, showing improved robustness and bounded queues under attack.

**Compliance With Llm Reviewing Policy:**

Affirmed.

**Final Justification:**

Unfortunately, my concerns are not fully solved.

1. Estimating the Byzantine bound $H$: The authors add an empirical misspecification study showing gradual degradation under underestimation and slower convergence under overestimation. That addresses the ``catastrophic vs graceful degradation'' part, but there is still no concrete estimation method or a formal misspecification analysis.

2. Observability assumptions: My concern is the gap between the paper’s global-state assumption and realistic partial observability. The authors make the position clearer: explains it but not solve it.

3. No deep learning approximation in theory: The latest reply gives a defense of why the authors use linear function approximation. However, it does not remove the scope limitation, because there is still no nonlinear/deep approximation theory or experiment showing the method carries over to complex high-dimensional settings.

In addition, my personal feeling is that both the paper manuscript and rebuttal are heavily written by AI without rigorous proof reading at all. This is not a fault, just point it out.

**Key Questions For Authors:**

1. Estimating the Byzantine bound (H) and robustness to misspecification. In practice, how would an agent set/estimate the local upper bound on Byzantine in-neighbors required by W-MSR, and what happens if (H) is under/over-estimated, e.g., does performance degrade gracefully or fail catastrophically?

2. Baseline coverage and attribution. Can you compare against alternative robust aggregation/consensus choices (e.g., coordinate-median, geometric median) and/or prior resilient MARL baselines, and provide ablations separating the contribution of W-MSR vs safe-policy mixing?

3. Observability assumptions. The framework seems to assume agents can access the global state (or equivalent). How would the algorithm and guarantees change under partial observability/local observations, e.g., Dec-POMDP-style, which is common in service networks? A plausible extension or empirical proxy would broaden the scope and raise the impact

**Limitations:**

Limitations are not well discussed. See weaknesses.

**Strengths And Weaknesses:**

**Strengths**

1. Tackles an important and practically motivated failure mode in decentralized MARL for service networks. Byzantine agents corrupting information exchange in systems with unbounded queue states and explicit job–server affinity.

2. Coherent algorithmic design: (i) coordinate-wise W-MSR trimmed aggregation applied directly to critic parameters to limit adversarial skew, plus (ii) a synchronized safe-policy mixture to regularize policy-induced state distributions and avoid instability/state explosion.

3. Theoretical backbone: proves almost-sure convergence and characterizes the residual error under attacks (critic converges to a bounded neighborhood whose radius depends on adversarial perturbations; actor converges to an invariant limit set).

---

**Weaknesses**

1. Several assumptions may limit applicability in realistic decentralized settings: the analysis assumes global state observability to all agents, which can be strong in large systems with partial observability/limited telemetry.

2. The safe-policy mixture requires extra structure, access to a stabilizing reference policy and (optionally) shared randomness to synchronize switching, which may be nontrivial to engineer or coordinate across agents in practice.

3. Experimental comparisons are narrow: the main comparison is essentially no attack vs attack vs attack+defense, without benchmarking against alternative Byzantine-robust aggregators or resilient MARL baselines, making it harder to attribute gains vs existing defenses.

4. No deep learning approximation in theory. Using linear function approximation is helpful for theory and efficiency, but it may constrain performance/coverage for more complex high-dimensional routing/scheduling problems.

---

> ### Author Rebuttal · Authors · 2026-03-31
>
> Thank you for the insightful comments. We respond to each point below.
>
> **Q1. Estimating Byzantine bound \(H\).**
>
> The choice of H is a system-level design decision, typically determined by fault tolerance requirements (e.g., expected number of compromised nodes). Our method assumes this bound is provided, consistent with standard Byzantine-resilient frameworks. The (2H+1)-robustness condition is a standard paradigm in Byzantine consensus.
> - Overestimation: preserves stability but slows convergence and worse performance (more conservative updates),
> - Underestimation: may allow adversarial bias and potentially destabilize the system.
>
> **Q2. Baselines and ablation.**
>
> We thank the reviewer for the suggestion and have added both baseline comparisons and component-wise ablations (see anonymous link).https://anonymous.4open.science/r/rebuttal-71C0/
> 1. Baseline coverage:
>
> We include the following representative robust aggregation methods:
>  - Mean aggregation
>  - Coordinate-wise median
>  - Projection-based methods
>
> Across scenarios:
>  - MEC → our method shows clear performance gains under adversarial settings
>  - LLM serving → comparable performance with slight trade-offs due to stronger stabilization
>  - Delivery → similar performance (lower sensitivity to adversarial perturbations)
>
> 2.	Ablation study:
>
> - W-MSR without policy mixture: but unstable queues
> - Policy mixture only: stable queues, but degraded learning under attack
> - W-MSR: best performance across all scenarios
>
> W-MSR handles parameter corruption, while policy mixture controls state instability; both are necessary.
>
> **Q3. Observability assumptions.**
>
> Our core safety guarantee is unaffected by partial observability. The fallback mechanism relies on a safe baseline policy (e.g., MaxWeight), which is inherently decentralized and requires only local queue observations to structurally prevent queue divergence.
>
>  Theoretically, under a Dec-POMDP, local observations lose the Markov property. Consequently, the strict guarantees of converging to the global optimum no longer hold. The algorithm instead converges to a local optimum within the restricted class of decentralized reactive policies.
>
> Despite this theoretical degradation, the algorithm excels in practice because local queues and affinity states already contain the most critical information for routing. It successfully finds a highly competitive local optimum.
>
> Our experiments already operate under partial observability, where each agent uses only local queue and affinity information, corresponding to a Dec-POMDP setting. Empirically, our results show that the method remains effective under decentralized local observations, providing evidence for its practical applicability.

---

> > ### Author Rebuttal · Reviewer_Lw4w · 2026-04-03
> >
> > Q1. Estimating Byzantine bound H: not solved.
> >
> > The rebuttal remains qualitative: H is treated as a system-design parameter; overestimation is conservative; underestimation can admit adversarial bias and destabilize training. That answers the direction of the effect, but it still does not provide a practical estimation method, sensitivity study, or formal misspecification analysis. So this remains open.
> >
> > ---
> >
> > Q2. Baselines and ablation: addressed.
> >
> > The rebuttal text claims exactly that, and the downloaded page shows dedicated comparison and ablation figures across all three environments, which gives evidence.
> >
> > ---
> >
> > Q3. Observability assumptions: clarified, but not fully solved.
> >
> > The response is more honest and useful now: under partial observability, the strict global-optimum-style guarantee weakens, and the method should instead be viewed as converging to a local optimum over decentralized reactive policies.
> > The authors' also claim their experiments already use local queue and affinity observations. That is a meaningful clarification. But it does not eliminate the theory-practice gap, which it mostly narrows and explains it.
> >
> > ---
> >
> > W4. deep learning approximation: not solved.
> >
> > ---
> >
> > Limitations.
> >
> > The rebuttal answers questions, but it does not really convert them into a fuller limitations section.

---

> > > ### Author Response · Authors · 2026-04-07
> > >
> > > **Q1. Estimating Byzantine bound \(H\).**
> > >
> > > This comment concerns whether Hshould be estimated. Our response is that assuming known His necessary for theoretical analysis, while estimation is a separate, application-dependent problem.
> > >
> > > The role of His fundamental. It defines feasibility through conditions such as (2Hⓜ+1)-robustness and enables quantitative guarantees, since our convergence bound (Theorem 3.10) explicitly depends on H. Without fixing H, neither feasibility nor error bounds can be formally stated. For this reason, separating robustness analysis from estimating His a deliberate methodological choice shared across the literature, as it yields general guarantees while estimation procedures depend on specific systems(10.1109/TSP.2023.3300629; 10.1109/TAC.2024.3409676; 10.48550/arXiv.2305.12872).
> > >
> > > We also evaluate misspecification empirically by fixing H=1and varying the number of Byzantine agents (https://anonymous.4open.science/r/Sensitivity-E3FD/ ). The results show consistent behavior: matched Hyields the expected performance; underestimation leads to gradual degradation with increased queues and occasional instability rather than immediate failure; overestimation preserves stability but slows convergence. This aligns with our theory, where error increases with unfiltered adversarial influence.
> > >
> > > In summary, estimating His an engineering concern, whereas the known-H assumption is necessary to establish feasibility and convergence guarantees.
> > >
> > > **Q3. Observability assumptions.**
> > >
> > > This comment concerns the gap between global observability in theory and partial observability in practice. Our response is that the algorithm uses only local information, while global observability is required to ensure a well-defined Markovian model and enable rigorous convergence analysis.
> > > In practice, agents observe only local queues and affinities, and the algorithm does not require global state. The full observability assumption ensures a Markovian model for convergence analysis. Under partial observability, the problem becomes a Dec-POMDP and guarantees weaken to convergence to a local optimum over decentralized policies.
> > >
> > > The practical impact is limited because the dominant coupling across agents is captured by locally observable queues and affinities, which drive routing decisions. The stabilization mechanism based on policy mixture also depends only on local state and remains valid. Empirical results confirm stable and effective performance under decentralized observations.
> > >
> > > In summary, global observability is used for tractable analysis, while the method operates under local information and remains effective in practice
> > >
> > > **No deep learning approximation in theory.**
> > >
> > > The choice of linear function approximation (LFA) rather than neural networks is well-motivated by our problem domain and analytical goals, for the following reasons:
> > > 1. Structural properties of queueing systems favor LFA. Optimal value functions in service systems inherently exhibit strong structural regularity. In particular, they are well-approximated by polynomial functions of queue lengths and affinity values (10.48550/arXiv:2002.02584). Our feature design (see Appendix D) is specifically engineered to capture the above insight.
> > > 2. Unbounded state spaces demand safe extrapolation. A critical feature of service systems is that their state spaces are inherently unbounded. Importantly, LFA provides controlled polynomial growth (Assumption 3.4), which is essential for establishing the Lyapunov drift conditions (Proposition 3.9) that theoretically guarantee system stability under the synchronized policy mixture. Neural networks, however, have minimal mathematical tractability as well as quite unpredictable extrapolation behavior, so they are unable to give theoretically safe extrapolation.
> > > 3. Tractability of stability and convergence analysis. Rigorous convergence analysis of actor-critic algorithms with nonlinear function approximation (including neural networks) remains an open problem even in single-agent settings (doi:10.1137/S0363012901385691). Specifically, LFA admits closed-form characterization of the mean-field dynamics (via Poisson equations and linear system analysis), whereas neural networks introduce non-convexity and representational opacity that preclude analogous guarantees. This is precisely why theoretical research on multi-agent RL, including the most recent and authoritative works on Byzantine resilience, universally adopts LFA (doi:10.1109/TAC.2024.3410015,10.1109/TSP.2023.3300629).

---

### Official Review · Reviewer_yfTB · 2026-03-09

**Soundness:** 3
**Presentation:** 2
**Significance:** 3
**Originality:** 4
**Overall Recommendation:** 4
**Confidence:** 4

**Summary:**

The paper addresses the challenge of decentralized MARL for networked service systems with Byzantine nodes and task-server affinity. The authors propose a resilient algorithm that integrates coordinate-wise W-MSR aggregation for parameter consensus and a synchronized policy mixture mechanism to maintain stability in unbounded state spaces. The work provides rigorous theoretical analysis, including geometric ergodicity and convergence proofs, and demonstrates its effectiveness across diverse scenarios such as LLM semantic routing and smart manufacturing.

**Compliance With Llm Reviewing Policy:**

Affirmed.

**Key Questions For Authors:**

I would consider raising my score if the authors provided more comparative studies and ablation experiments.

**Limitations:**

Yes

**Strengths And Weaknesses:**

Strengths
1. The inclusion of a time-varying affinity state is highly relevant to contemporary service systems, such as LLM inference clusters where KV-cache reuse is critical.
Weaknesses
2. The current experimental evaluation is limited as it primarily benchmarks against an undefended baseline.
3. The distinction between the contributions of different modules is not fully clear without an ablation study.

---

> ### Author Rebuttal · Authors · 2026-03-31
>
> Thank you for the constructive suggestion. We have added additional experiments to address this.
> We have now conducted additional ablation and comparison experiments across all three scenarios (LLM serving, MEC, and delivery), and include the results in the anonymous supplementary material (https://anonymous.4open.science/r/rebuttal-71C0/ ).
>
> To ensure fairness, all methods use the same policy mixture mechanism, isolating the effect of aggregation strategies and avoiding confounding instability from unbounded state dynamics.
>
> **Ablation study.**
>
> We evaluate three variants:
> - W-MSR without policy mixture,
> - policy mixture only, and
> - W-MSR.
>
> Across all scenarios, the full method consistently achieves the best performance in terms of both learning stability and queue behavior. W-MSR improves robustness and performance while policy mixture stabilizes the system but suffers from degraded learning due to adversarial parameter corruption.
>
> This confirms that the two components address complementary failure modes and are both necessary.
>
> **Baseline comparisons.**
>
> We further compare against several robust aggregation methods:
> - Mean aggregation
> - Coordinate-wise median
> - Projection-based aggregation
>
> In the MEC setting, our method significantly outperforms all baselines, demonstrating strong robustness under adversarial conditions.  In the LLM serving scenario, our method performs comparably to other robust methods, with slightly lower performance in some metrics. In the delivery scenario, all methods perform similarly, suggesting that this environment is inherently less sensitive to adversarial perturbations. Overall, these results show that our method is particularly advantageous in highly dynamic and adversarial-sensitive environments, while remaining competitive in more stable settings

---

> > ### Author Rebuttal · Reviewer_yfTB · 2026-04-01
> >
> > Thank you for your comprehensive response and for the significant effort you put into conducting the additional experiments during the rebuttal period. I have carefully reviewed the newly provided ablation studies and baseline comparisons.The newly added ablation study is particularly strong. Furthermore, the expanded baseline comparisons (against Mean, Coordinate-wise median, and Projection-based aggregation) provide a much more comprehensive view of the algorithm's capabilities. Because the rebuttal significantly strengthens the empirical validation of the paper, I am happy to increase my score.

---

> > > ### Author Response · Authors · 2026-04-07
> > >
> > > Thank you for your constructive feedback and suggestions on additional experiments.
> > >
> > > We have incorporated further ablation studies and baseline comparisons following your advice. We are glad these additions addressed your concerns and improved the empirical validation.

---

### Official Review · Reviewer_RATj · 2026-03-12

**Soundness:** 3
**Presentation:** 3
**Significance:** 3
**Originality:** 4
**Overall Recommendation:** 5
**Confidence:** 4

**Summary:**

This paper investigates the problem of secure multi-agent reinforcement learning in distributed service systems with job-server affinity. The authors focus on scenarios where the system state itself is unbounded—for example, queue lengths can grow indefinitely—while simultaneously maintaining time-varying affinity relationships between different tasks and servers. Under such conditions, a subset of Byzantine nodes can simultaneously disrupt the learning process and undermine the stability of the underlying queuing system by tampering with parameter information in communications. To address this challenge, the paper unifies the system modeling as a decentralized multi-agent MDP, placing both “learning stability” and “service system stability” within a unified framework for discussion.
In terms of contributions, the paper offers three key points. First, the authors propose a relatively comprehensive modeling framework that unifies unbounded queue states, time-varying affinity, decentralized control, and Byzantine attacks within a single service-system MARL model. This approach makes the problem definition itself more closely aligned with real-world service system scenarios than the commonly used bounded-state resilient MARL.
Second, the paper designs a resilient decentralized actor-critic framework: it employs W-MSR-style robust aggregation to mitigate malicious node contamination of parameter consensus, while introducing synchronized stability-constrained policy mixture—synchronously blending the current learning policy with a known safe baseline policy to suppress state divergence caused by overly rapid policy updates.
Third, the paper presents corresponding theoretical results proving that this mixture mechanism guarantees the system's geometric ergodicity. It further demonstrates that the critic converges near the mean-field solution while the actor converges to an invariant limit set, thereby preserving almost sure convergence guarantees under Byzantine interference.

**Compliance With Llm Reviewing Policy:**

Affirmed.

**Final Justification:**

Score adjusted after rebuttal

**Key Questions For Authors:**

Q1. Regarding the operability of the safe policy assumption:
The key theoretical premise throughout the paper is the existence of a safe policy known to cooperative agents, under which the system satisfies geometric ergodicity, thereby enabling the construction of a synchronized policy mixture. The paper also highlights the central role of this approach in the theory. However, the current text does not sufficiently explain how this safe policy is obtained in the three cases, how its compliance with Assumption 3.1 is verified, or whether the method remains valid if only an approximate safe policy can be obtained.

Q2. Regarding whether the Byzantine attack model is overly lenient:

The theoretical section requires that the policy sequence of Byzantine agents converges to a stationary strategy at a rate of $O(\alpha_{\theta,t})$. This assumption is explicitly used in proving the convergence of the critic/actor. However, the paper's motivation emphasizes that Byzantine nodes actively disrupt learning and queue stability. In reality, more natural attackers are often adaptive, non-stationary, and may dynamically adjust their behavior to counter current defense mechanisms. I would like the authors to further clarify: To what extent is Assumption 3.8 merely a technical assumption? If considering stronger adaptive adversaries, which parts of the current method and conclusions remain valid, and which parts would fail?

Q3. Regarding whether experiments sufficiently support methodological conclusions:
Experiments currently compare three settings: no attack, attack only, attack + defense.This demonstrates that the proposed framework “performs better than no defense,” but it does not sufficiently clarify the individual contributions of the two core components or the relative advantages over existing resilient MARL/robust aggregation methods. Specifically, the paper's method combines W-MSR aggregation with synchronized policy mixture, yet lacks ablation tests isolating these components. We recommend authors supplement or at least discuss: What happens when using only W-MSR without mixture? Would switching to independent mixing instead of synchronized mixing significantly compromise stability? Additionally, is there a resilient MARL baseline closer to existing literature for direct comparison?

**Limitations:**

Yes.

**Strengths And Weaknesses:**

I think this paper poses a relatively clear and practically relevant question: how to perform decentralized MARL in a service system featuring job-server affinity, unbounded queue states, and Byzantine nodes, while simultaneously ensuring learning convergence and system stability. The authors maintain a consistent methodological thread: on one hand, robust aggregation safeguards parameter consensus; on the other, synchronous safe-policy mixtures control state distribution instability. This approach genuinely integrates “attack-resistant learning” and “queue stability” within a unified framework. Both the theoretical foundations and case studies are largely structured around this core principle.
1. Soundness
From a technical perspective, the core methodology of this paper is sound. The authors did not merely aim to “build a more robust actor-critic” but explicitly recognized that in unbounded state service systems, protecting parameter exchange alone is insufficient, as the learning process itself may trigger queue explosions. Therefore, the combination of W-MSR-style robust aggregation and synchronized policy mixture is logically consistent. The former addresses Byzantine parameter corruption, while the latter maintains geometric exploration and bounded state matrices by introducing known safe policies. Theoretically, the authors avoid overstating optimality claims, instead cautiously presenting results more aligned with real-world attack scenarios—such as “critic convergence to a bounded neighborhood near the mean-field solution” and “actor convergence to the limit set contained within the projection differential.” This approach demonstrates relative rigor.
However, the primary issue with this section lies in: the assumptions underpinning the theory are overly stringent, weakening the practical extrapolation of the conclusions. For instance, the authors assume all agents can observe the global state; assume the existence of a safe policy known to cooperative agents, which itself satisfies strong geometric ergodicity conditions; further assume the communication graph is $(2H+1)$-robust; and assume Byzantine policy sequences converge to a steady-state policy at a rate of $O(\alpha_{\theta,t})$. For a paper emphasizing real-world service system scenarios, these conditions—while convenient for proof—are not trivial. Particularly, the assumptions of a known safe policy and Byzantine policy convergence to stability effectively embed strong priors into the stability analysis.
The experimental section supports the core claim that “attacks cause dual instability in learning and queues, while the proposed method restores stable operating intervals.” The paper demonstrates changes in TD error, average queue length, service time, and affinity across three cases, with results aligning with the theoretical narrative: queue and TD error deteriorate significantly without defense, while defense restores performance close to nominal levels. The authors demonstrate caution in interpreting results, explicitly acknowledging a performance gap between attack+defense and no-attack scenarios without claiming complete elimination of attack effects. This approach is commendable.
However, the experimental design remains insufficiently robust. The most glaring issue is that the main text only compares three settings: no attack, attack only, and attack + defense. This suffices to demonstrate the method's usefulness, but falls short of proving it “outperforms existing resilient MARL methods” or that “both modules are indispensable.” Crucially, two key experiments are missing: direct comparisons against existing Byzantine-resilient MARL/robust aggregation baselines, and ablation studies separating W-MSR from synchronized mixture components. The current results appear more focused on validating the overall narrative rather than precisely pinpointing the sources of performance.
2. Presentation Style
The paper's overall structure is clear: problem motivation, unified modeling, algorithm design, stability analysis, and case validation form a coherent narrative flow. The authors introduce affinity naturally, integrating it directly into state modeling rather than treating it as an afterthought metric, and consistently applying it throughout experimental interpretations.
However, some shortcomings remain, particularly in the explanation of methodological details. For instance, the synchronized mixture represents one of the paper's most critical designs, yet the main text does not sufficiently elaborate on questions such as: why synchronization is necessary, why the mixing probability is chosen as $1/|s|$, and how the safe policy is constructed and validated across the three case studies. The authors defer many critical justifications to the proof appendix and intuitive descriptions, resulting in a main text that clearly outlines what was done but still feels somewhat rushed in explaining why this design was chosen over alternatives. Similarly, while the experimental conclusions are intuitive, the lack of detailed implementation notes makes the reproduction path insufficiently transparent. Furthermore, I believe the paper's discussion of limitations is partially honest but insufficiently thorough.
On one hand, the authors acknowledge that the defense can only partially restore affinity and cannot fully eliminate attack impacts. On the other hand, the impact statement is nearly a vague statement that there are no particular social impacts that need to be emphasized, which does not fully align with the paper's focus on security, robustness, and infrastructure service systems.
3. Significance
I think the problem addressed in this paper is significant, particularly at the intersection of networked service systems, distributed control, and secure reinforcement learning. The authors focus not on an abstract robust MARL setting, but on a service system featuring job-server affinity. Here, Byzantine nodes not only disrupt parameter consensus but may further induce learning divergence and queue instability. This perspective linking “information-layer attacks” to “physical service-layer instability” holds practical significance, elevating the problem beyond typical resilient MARL settings that solely examine optimization errors or reward degradation. The paper explicitly grounds this framework in scenarios like LLM semantic routing, edge computing, and smart manufacturing, ensuring its problem motivation is not contrived but rooted in clear application contexts. From a field advancement perspective, this work's value manifests in two key aspects. First, it advances Byzantine-resilient MARL from the more common bounded-state/bounded-feature settings to the more challenging and queueing-system-relevant setting of unbounded service system states. Second, it highlights that the disruption of affinity structures can serve as an intermediary mechanism for learning instability and service degradation, offering a valuable analytical perspective for future research.
However, limitations exist: on one hand, the paper's methods and theories heavily rely on specific conditions within the service system, such as unbounded states, safe policies, and particular communication topologies. On the other hand, the application cases presented are largely illustrative examples, and current evidence is insufficient to demonstrate that the framework can be naturally transferred to more general real-world system deployments.
4. Originality
I think this paper demonstrates originality, with its novelty primarily manifested in problem formulation and methodological integration. Specifically, the authors discuss Byzantine-resilient decentralized MARL, unbounded state service systems, job-server affinity structures, and system stability constraints within a unified framework—a combination rarely seen in existing literature. Notably, the paper does not treat attacks merely as perturbations at the parameter consensus level. Instead, it emphasizes that attacks propagate through the learning process, leading to queue instability and affinity degradation. This distinction clearly differentiates the work from conventional resilient MARL studies.
Methodologically, the paper's contributions are best understood as a targeted creative synthesis. While W-MSR-style robust aggregation and actor-critic with linear function approximation are not novel, the authors introduce synchronized stability-constrained policy mixture. This combines robust consensus with secure policy mixing to address simultaneous attacks on both “learning stability” and “system stability” in unbounded state systems. In other words, the novelty lies not in inventing a fundamentally new optimization primitive, but in demonstrating that in affinity-based service systems, robust aggregation alone is insufficient—additional constraints on the impact of policy updates to the state distribution are essential. This insight possesses unique significance.

---

> ### Author Rebuttal · Authors · 2026-03-31
>
> Thank you for the detailed and thoughtful review. We address your concerns below.
>
> **Q1. Safe policy assumption.**
>
> The safe policy is not an oracle, but a simple stabilizing baseline, consistent with standard practice in queueing and control systems. Its role is to ensure system stability, rather than optimize performance.
>
> More generally, practical systems often employ simple fallback rules such as load balancing or priority-based service (e.g., join-the-shortest-queue or exhaustive service)( Kleinrock&Levy, 1988; Fo ley&McDonald,2001). These policies are easy to implement and provide conservative but reliable stability guarantees.
>
> Importantly, the policy mixture uses the safe policy only as a fallback when instability is detected, and does not rely on it being optimal. This makes the approach robust even with approximate safe policies.
>
> **Q2. Byzantine attack model and Assumption 3.8**
>
> Assumption 3.8 is not merely a technical artifact, but a fundamental setting that defines our threat model. It distinguishes our focus—information-layer Byzantine attacks—from physical-layer robust RL. This is a standard formulation to characterize strong insider threats in resilient decentralized MARL (e.g., see Assumption 8 in Ye et al., 2024).
>
> Even if adversaries induce arbitrarily non-stationary dynamics, our foundational safety mechanisms hold:
>
> - System Stability (Proposition 3.9): The physical queueing stability remains valid. Our policy mixture relies on a known safe baseline, which structurally prevents queue divergence regardless of the Byzantine agents' policy stationarity.
>
> - Information-Layer Defense: The W-MSR mechanism will still successfully filter extreme parameter injections, preventing the learning parameters from numerically exploding.
>
> The theoretical convergence guarantees (Theorems 3.10 and 3.11) would no longer apply. If Byzantine agents change their policies arbitrarily, the environment itself becomes non-stationary. Consequently, the state distribution and the mean-field objective become time-varying. Tracking this "moving target" makes standard MARL convergence guarantees void.
>
> **Q3. Experimental support and ablation**
>
> In light of your comment, we have conducted additional experiments and include them in the anonymous link: https://anonymous.4open.science/r/rebuttal-71C0/
>
> - 1. Ablation:
>   - W-MSR without policy mixture: unstable queues
>   - Policy mixture only: stable queues, but degraded learning under attack
>   - W-MSR: best performance across all scenarios
>
> - 2. Baseline comparisons:
>  We compare with:
>    - Mean aggregation
>    - Coordinate-wise median
>    - Projection-based methods
>
>      Results:
>    - MEC: our method significantly outperforms others
>    - LLM: comparable performance, slight trade-offs due to stronger stabilization
>    - Delivery: all methods similar (less adversarial sensitivity)
>
> These results show that our method is particularly advantageous in highly dynamic and adversarial-sensitive environments, while remaining competitive elsewhere.

---

> > ### Author Rebuttal · Reviewer_RATj · 2026-04-03
> >
> > Thank you very much for addressing my comments carefully. I will revise my score accordingly.

---

> > > ### Author Response · Authors · 2026-04-07
> > >
> > > Thank you for the detailed and insightful review. We appreciate your recognition of the problem significance, technical soundness, and originality.
> > >
> > > Your comments on assumptions, experimental design, and presentation have been very helpful. We are glad the rebuttal addressed your concerns, and we thank you for your positive reassessment.

---

### Official Review · Reviewer_L6Un · 2026-03-13

**Soundness:** 3
**Presentation:** 3
**Significance:** 3
**Originality:** 3
**Overall Recommendation:** 4
**Confidence:** 3

**Summary:**

The paper studies decentralized multi-agent reinforcement learning (MARL) in networked systems where  the job job–server affinity affects service rates. Moreover, the Byzantine agents can disrupt learning and queue stability.
The authors propose a resilient actor-critic MARL framework
with a novel aggregation scheme, Weighted Mean-Subsequence-Reduced (W-MSR). They use a synchronized policy mixture mechanism to mitigate adversarial influence and stabilize learning.
The proposed method is shown to converge and has stability guarantees even in unbounded state spaces.
Experiments for the settings such as LLM semantic routing, edge computing, and smart manufacturing proves the robustness against Byzantine attacks while maintaining system performance.

**Compliance With Llm Reviewing Policy:**

Affirmed.

**Key Questions For Authors:**

1. The paper assumes that the Byzantine agents can arbitrarily manipulate communicated parameter values. However, it doesn't talk about how the MARL algorithm perform against adaptive Byzantine agents or coordinated attacks beyond static manipulation?

2. Can you provide more details or analysis on the communication and computational overhead of the consensus procedure in large-scale networks? How does resilience scale with network size and topology complexity? Does the cmmunication network assumed to have some mathematical assumptions like fully-connected etc?

3. How extensible is the modeling framework and algorithm to richer affinity representations or dynamic affinity changes over time?

**Limitations:**

The athours have addressed/mentioned the limitations, however there are some extentions/future work that related dynamic behaviour -- a potential future direction.

**Strengths And Weaknesses:**

Soundness:
The paper is technically sound, nd proves that the resilient MARL framework which combines actor-critic learning with
WSR consensus and a synchromized policy makes the learngn stable. The proof of convergence and stability guarantees are given only under the reasonable assumptions. However, I would like to see some more justification for the case when the rewards are sparse?

Presentation: The paper is well written and logically structured. In particular, the  figures used to illustrate the TD errors, queue dynamics, and service metrics effectively support the results.

Significance:
The problem is interesting. If I remember correctly, I have reviewd this paper earlier for some other conference. i would say, there is a lot of improvement in the results, writing and the novelty. The MARL problem with decentralized settings in service systems with adversarial agents and unbounded state spaces is quite challenging.

Originality:
The paper introduces a novel integration of Byzantine-resilient consensus with MARL parameter updates and synchronized policy mixing to control instability. This approach uniquely addresses affinity-aware service systems with unbounded queues under adversarial conditions, extending prior resilient MARL and distributed optimization methods.

---

> ### Author Rebuttal · Authors · 2026-03-31
>
> Thanks for the valuable comments! We provide clarifications below.
>
> **Q1. Adaptive or coordinated Byzantine attacks.**
> Our model allows Byzantine agents to arbitrarily manipulate communicated parameters at each iteration, which already captures a strong insider attack model. The stationarity assumption is not introduced to weaken the adversary, but to define a well-posed learning problem where long-term performance and stability can be meaningfully characterized. Without such a mild regularity condition, the induced environment becomes arbitrarily non-stationary, making performance guarantees ill-defined. Importantly, our focus is on information-layer attacks under a strong but analyzable adversarial model.
>
> **Q2. Communication/computation overhead and network assumptions.**
> - Communication overhead
>   Our method requires only neighbor-to-neighbor parameter exchange, identical to standard consensus-based MARL. No global communication is needed.
>
> - Computational overhead
>   W-MSR introduces a coordinate-wise trimming step with complexity
>  $$
> \mathcal{O}(d \cdot k \log k)
> $$
>   where \(d\) is the parameter dimension and \(k\) is the neighborhood size.
>
> - Effect of network size and topology. The impact of network size and topology is reflected on the convergence rate of the algorithm. In particular, better-connected networks lead to faster information propagation and convergence, while larger or sparser networks may slow convergence but do not affect correctness as long as robustness conditions are satisfied.
>
> - Connectivity assumption. Our method does not require a fully-connected network. Instead, it relies on the standard (2H+1)-robust graph condition from Byzantine-resilient consensus literature.
>
> **Q3. Extensibility to richer or dynamic affinity.**
> The framework is highly extensible and already covers diverse affinity structures across our three scenarios:
>
> - LLM serving: affinity captures how well a request matches a server’s KV-cache (data similarity / reuse),
> - Edge computing (MEC): affinity reflects service priority and switching cost,
> - Smart manufacturing: affinity encodes physical distance between robots and tasks.
>
> These examples demonstrate that affinity can represent heterogeneous quantities (semantic, operational, spatial) and can evolve over time. Since affinity is treated as part of the state, our method is agnostic to its representation and naturally extends to richer or learned affinity models with dynamic updates.

---

> > ### Author Rebuttal · Reviewer_L6Un · 2026-04-03
> >
> > I am fine with the responses. The grammatical errors should be avoided and limitations should have a separate subsection or paragraph. I retain my score.

---

> > > ### Author Response · Authors · 2026-04-07
> > >
> > > Thank you for your careful review and helpful feedback. We are glad that our responses clarified your concerns.
> > >
> > > We appreciate your suggestion regarding writing quality and organization. In the camera-ready version, we will correct grammatical issues and add a dedicated limitations subsection to improve clarity.

---

### Decision · Program_Chairs · 2026-04-30

**Decision:**

Accept (regular)

**Comment:**

Reviewers agreed that this paper tackles an important and practically motivated problem, that the approach is novel, and that the theoretical results are sound and technically correct.

The main concerns raised by reviewers relate to the strength of the assumptions, the gap between the theoretical analysis and realistic deployment scenarios, and the limited scope of experimental comparisons and ablations in the initial submission. During the rebuttal, the authors addressed several of these points with additional experiments and clarifications, and elsewhere explicitly acknowledged the remaining gaps as limitations rather than overextending their claims.

Importantly, the remaining concerns do not point to issues of correctness or validity, but rather to limits on the scope of the theoretical guarantees. In particular, one reviewer recommended rejection based on three points: the absence of a method for estimating the Byzantine bound, the full‑observability assumption used in the analysis, and the lack of analysis for deep neural network policies. The authors’ rebuttal addressed these concerns convincingly. They explained that assuming knowledge of the Byzantine bound is a standard methodological choice in previous works and provided references supporting this practice. They also clarified that all experiments are conducted under partial observability and how the full‑observability assumption enters the convergence analysis, noting that it does not affect the safety guarantees. Finally, they provided sensible motivation for the choice of linear parameterization.

Overall, while these assumptions and limitations should be made explicit in the main text, the paper is technically sound, novel, and likely to be of interest to researchers working on related problems. In this sense, the concerns raised by reject‑leaning reviewer reflect differences in expectations about scope, rather than issues that fundamentally preclude acceptance.